# Learning to solve Class-Constrained Bin Packing Problems via Encoder-Decoder model

**Hanni Cheng**  **Ya Cong**[*]  **Weihao Jiang**  **Shiliang Pu**

Hikvision Research Institute, Hangzhou 310051, China
`{chenghanni, congya, jiangweihao5, pushiliang.hri}@hikvision.com`

## Abstract

Neural methods have shown significant merit in solving combinatorial optimization (CO) problems, including the Bin Packing Problem (BPP). However, most existing ML-based approaches focus on geometric BPP like 3DBPP, neglecting complex vector BPP. In this study, we introduce a vector BPP variant called Class-Constrained Bin Packing Problem (CCBPP), dealing with items of both classes and sizes, and the objective is to pack the items in the least amount of bins respecting the bin capacity and the number of different classes that it can hold. To enhance the efficiency and practicality of solving CCBPP, we propose a learning-based Encoder-Decoder Model. The Encoder employs a Graph Convolution Network (GCN) to generate a heat-map, representing probabilities of different items packing together. The Decoder decodes and fine-tunes the solution through Cluster Decode and Active Search methods, thereby producing high-quality solutions for CCBPP instances. Extensive experiments demonstrate that our proposed method consistently yields high-quality solutions for various kinds of CCBPP with a very small gap from the optimal. Moreover, our Encoder-Decoder Model also shows promising performance on one practical application of CCBPP, the *Manufacturing Order Consolidation Problem* (OCP).

## 1 Introduction

The Bin Packing Problem (BPP) is a classic combinatorial optimization (CO) problem with numerous practical applications in various industries, such as manufacturing, logistics, and resource allocation. Approximate heuristics methods and the exact algorithms have traditionally been used to find good solutions, but they may either lack solving efficiency or generalizability. Recently, a variety of machine learning (ML) based methods (Kwon et al., 2020; Ma et al., 2021; Cheng et al., 2023; Jiang et al., 2021; Zhao et al., 2021a;b; Zhang et al., 2020; 2023) have been proved to be effective means to deal with complex CO problems including BPP with appealing advantages. These learning-based methods outperform both exact solvers and approximate heuristics in terms of solution efficiency as well as generalization ability.

There are primarily two generalizations of bin packing, geometric bin packing and vector bin packing (Christensen et al., 2016). The majority of the learning-based BPP solvers focus on geometric BPP like offline and online 3D bin packing problem (3DBPP), which usually refers to packing a set of cuboid-shaped items along with $x$, $y$, and $z$ axes respectively, into the minimum number of bins in an axis-aligned fashion. These methods usually leverage Attention Networks or Convolution Neural Networks as the Encoder to better represent the space constraints and learn to construct a solution sequentially via reinforcement learning (RL) which learns the solver from the generated packing sequences (Duan et al., 2018; Zhang et al., 2021; Zhao et al., 2021a;b; Jiang et al., 2021). On the other hand, vector BPP with multiple properties and complex constraints have received limited attention in the neural CO problems field thus far. A typical example of vector BPP is the Class-Constrained Bin Packing Problem (CCBPP), which deals with items of classes and sizes, and the objective is to pack the items in the least amount of bins respecting the bin capacity and the number of different classes that it can hold.

---

[*]Corresponding Author

Figure 1: The pipeline of the Encoder-Decoder Model.

Our work aims to learn a novel learning-based model that can effectively solve various kinds of CCBPP that have multiple properties and complex constraints. In the context of CCBPP, only the packing sequences need to be noted, without the position and orientation that need to be considered in 3DBPP. However, an optimal packing result always corresponds to numerous packing sequences since shuffling the order of items within each bin does not affect the packing solution but will directly alter the packing sequence. Thus, it brings complexity when relying solely on learning from packing sequences. To overcome these challenges, we introduce the connection matrix as a label to represent whether different items are packed together in the optimal result. If two items are packed together, the corresponding value in the matrix is 1, otherwise, its value is 0. The connection matrix is providing richer information compared to the generated packing sequences.

In this paper, we propose a learning-based Encoder-Decoder Model to get an approximate solution of CCBPP, as illustrated in Fig. 1. Our approach involves the construction of a graph based on item information, as the sizes and classes of the items can be effectively represented through a graph structure. The training labels are defined as the true connection matrix depicting relationships among various items. GCN is leveraged to train an Encoder so that the connectivity matrix with probabilities, also be treated as a heat-map, of the test instance can be generated. After the Encoder is well-trained, we introduce Cluster Decode algorithm to decode the heat-map matrix to an item sequence. Moreover, the test instances may suffer from significant differences in the distribution from training samples, which may lead to relatively poor performance. Therefore, Active Search technique is utilized during the decoding to fine-tune the final solution, exploiting the instance-specific characteristics to ensure solution quality. Our Model is different from conventional Encoder-Decoder structures to solve CO problems (Vinyals et al., 2015; Kool et al., 2019; Li et al., 2019) that the Encoder and the Decoder are two separated parts with different purposes, which makes the Encoder easier to learn from the connection matrix, and it will be more flexible to be applied in real-word scenarios. Overall, the main contributions are summarized as follows:

- We give a clear definition of CCBPP, and introduce a practical application of CCBPP, the *Manufacturing Order Consolidation Problem* (OCP) in detail. We show that CCBPP and OCP can be formulated as special variants of vector BPP.

- We introduce a novel model to solve packing problems by first **encode** the properties and dominant constraints by GCN to generate the connectivity probabilities of different items, and **decode** the best solution according to the particular attributes of one instance. As far as we know, we are the first to propose a learning-based method to solve complex vector BPP.

- We conduct extensive experiments on various settings of CCBPP synthetic datasets and real-world OCP datasets, and the results show that our algorithm can obtain better solutions than other benchmarks, while also spending less time.

## 2 RELATED WORK

**1DBPP** The 1D bin packing problem (1DBPP) is one of the most well-known problems in combinatorial optimization, and related research dates back to the 1960s (Kantorovich, 1960). Due to the NP-hard nature of 1DBPP, many heuristic approximation algorithms have been designed and their worst-case performance analysis has also been explored (Coffman et al., 1996). For example, the Next Fit (NF) algorithm, the First Fit (FF) algorithm, and the Best Fit (BF) algorithm are the most well-known greedy algorithms to solve 1DBPP.

**CCBPP**  CCBPP is one special variant of vector BPP. Many authors have studied the CCBPP and proposed various algorithms to solve it. Shachnai & Tamir (2001) introduced the CCBPP with applications to a data placement problem. Later on, Xavier & Miyazawa (2006) presented various approximation algorithms for the online and offline versions of CCBPP which is motivated by applications on video-on-demand systems. Shachnai & Tamir (2002) presented algorithms for the online CCBPP problem when all items have unit sizes. Epstein et al. (2010) followed up on the previous works that improved upon existing algorithms and offered a study of the approximation of the CCBPP. Lin et al. (2013) proposed a heuristic approach for the online version of this problem. Most of these works focus on a specific application of CCBPP, and they mainly provide the theoretical analysis of the approximation scheme, while our method is a learning-based algorithm on general CCBPP. Recently, Borges et al. (2020) presented a Branch-and-Price framework to solve the CCBPP with two different branching schemes, da Silva & Schouery (2023) proposed branch-and-cut-and-price framework to solve cutting stock problems including CCBPP. These two frameworks are exact algorithms, which are different from ours.

**Neural methods for BPP**  Learning-based methods to solve BPP mainly focus on geometric BPP like offline and online 3D bin packing problems. Duan et al. (2018) proposed a multitask framework based on Selected Learning to solve 3D flexible bin packing problem. Zhao et al. (2021a); Jiang et al. (2021) leveraged deep reinforcement learning (DRL) method to solve online 3DBPP. Later, Zhao et al. (2021b) tried to enhance the practical applicability of online 3DBPP via learning on packing configuration trees. For offline settings of BPP, Zhang et al. (2021) proposed a new end-to-end learning model based on self-attention-based encoding and DRL algorithms for BPP, and Zhu et al. (2021) proposed a data-driven tree search algorithm (DDTS) to tackle the large-scale offline 3DBPP.

## 3 PROBLEM DESCRIPTION

**Class-Constrained Bin Packing Problem**  In the Class-Constrained Bin Packing Problem, given positive integers $B$, $C$, $Q$ and a set $\{1, ..., N\}$ of items along with their size $s_i \in \mathbb{N}^+$ and class $c_i \in \{1, ..., Q\}$ of item $i$, one must partition the set of items so that the sum of the item sizes in each part is at most $B$ and the number of different classes in each part is at most $C$ while keeping the number of parts to a minimum. Thus, $B$ is the size of the bin, $C$ is the number of different classes that a bin can hold and $Q$ is the number of classes of all the items. The objective of CCBPP is to minimize the number of bins: $minimize \ \sum_{j=1}^{N} y_j$.

Single-class constrained BPP and multi-class constrained BPP are two kinds of general CCBPP. In single-class constrained BPP, one item belongs to only one class, *Data placement* (Golubchik et al., 2000) and *Production planning* (Davis et al., 1993) are typical single-class constrained versions of BPP. Multi-class version means one item belongs to multiple classes, *Co-Painting* (Peeters & Degraeve, 2004) and *Manufacturing Order Consolidation Problem* discussed below are typical applications of multi-class constrained BPP.

**Order Consolidation Problem**  The *Manufacturing Order Consolidation Problem* is a practical application of CCBPP in the domain of supply chain management which will also be discussed in this paper. Specifically, a manufacturing order (item) involves multiple components (classes) supplied by a component feeder. The feeder has a fix-size thus it can only provide limited KINDS of components in one production process. Various orders can be combined into a new order (packing into a bin) to be produced, as long as the total kinds of components required for their production do not exceed the feeder's supply capacity. The primary objective of OCP is to minimize the total amount of orders after the consolidation process, which can greatly save time for equipment switching and improve productivity. More details of OCP can be found in Appendix B.2.

## 4 METHODOLOGY

### 4.1 FRAMEWORK

The overview of our framework is illustrated in Fig. 1. The input data is the properties of items that should be packed, e.g. the classes and sizes of items in CCBPP or the size of components one order contains in OCP. Firstly, we construct a graph according to the variables and constraints of

the problems, and then the input graph is encoded by a well-defined Graph Convolution Network to generate a heat-map, which represents the probabilities of items packing together. Cluster Decode is then leveraged to decode the heat-map into a sequence format and output the final solution according to the characteristics of the particular sample with Active Search. In the following sections, we will introduce the **Encoder** and the **Decoder** in detail.

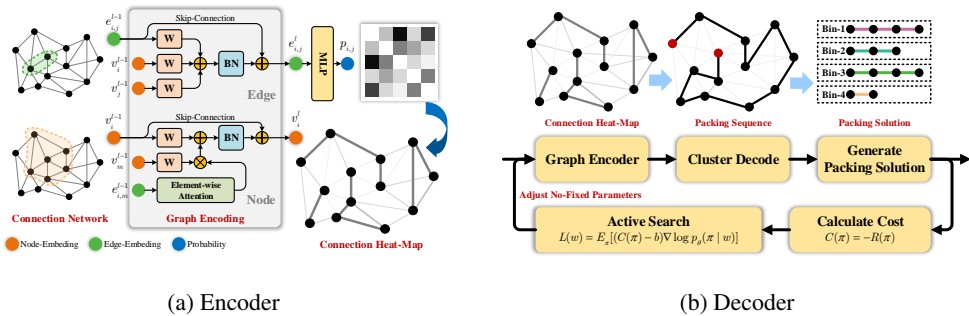

(a) Encoder        (b) Decoder

Figure 2: An illustration of the Encoder and the Decoder. The Encoder takes the well-constructed graph as input and outputs the heat-map which shows the probabilities of items packing together. The Decoder decodes the heat-map into a packing sequence, along with the Active Search to fine-tune the result according to the characteristics of the input instance.

## 4.2 ENCODER

In the domain of CCBPP, it is essential to acknowledge that various items may share the same class, implying relationships between them. Since graph is a structure amounting to a set of objects in which some pairs of the objects are in some sense "related" (Fraughnaugh, 1997), it can be leveraged to represent this problem where these items are the vertices and their class relationships are the edges. This approach enables us to harness the power of Graph Convolutional Network as our learning model, allowing to effectively capture the intricate patterns inherent in CCBPP. GCN is uniquely equipped to leverage both the attributes of vertices and the connections between them, making them a well-suited choice for addressing the complexities of this problem. The details of the Encoder are shown in Fig. 2a.

The network is trained by supervised learning and the output of the network is the heat-map. The structure of GCN and the training details will be presented in the following.

### 4.2.1 GRAPH CONVOLUTION NETWORK

We format the CCBPP as a full connected undirected graph $G = (V, E)$ containing the node set $V$ and edge set $E$.

In our GCN input, node attributes $x_v \in \mathbb{R}^d$ represent item properties, such as size and class. Edge attributes $x_e \in \mathbb{R}$ represent the relationships between items. Similar class affiliations suggest a tendency for items to be packed together, hence we use class cosine similarities as edge features. To account for items without shared classes, zero cosine similarities are replaced with a small constant $\epsilon$. The GCN output provides probabilities of two items being packed together. At the beginning, the node inputs $x_v$ and the edge inputs $x_e$ are linearly projected into different vectors, $v_i^0 \in \mathbb{R}^H$ and $e_{i,j}^0 \in \mathbb{R}^H$, respectively, $H$ is the feature dimension, $i \in V$ and $(i, j) \in E$.

$$v_i^0 = A_v x_v + b_v \tag{1}$$

$$e_{ij}^0 = A_e x_e + b_e \tag{2}$$

where $A_v \in \mathbb{R}^{H \times D}$ and $A_e \in \mathbb{R}^{H \times 1}$. Then the projected node and edge embeddings are fed into $L$ Graph Convolution layers. Let $v_i^l$ and $e_{i,j}^l$ denote the node feature vector and the edge feature vector respectively, the features at layer $l$ can be defined as follows:

$$attn_{i,j}^l = exp(W_0^l e_{i,j}^{l-1}) \oslash \sum_{(i,m) \in E^*} exp(W_0^l e_{i,m}^{l-1}) \qquad (3)$$

$$v_i^l = v_i^{l-1} + ReLU(BN(W_1^l v_i^{l-1} + \sum_{(i,m) \in E^*} attn_{i,j}^l \odot W_2^l v_j^{l-1})) \qquad (4)$$

$$e_{i,j}^l = e_{i,j}^{l-1} + ReLU(BN(W_3^l v_i^{l-1} + W_4^l v_j^{l-1} + W_5^l e_{i,j}^{l-1})) \qquad (5)$$

where $W \in \mathbb{R}^{H \times H}$, $\odot$ represents the element-wise multiplication and $\oslash$ represents the element-wise division. Skip-connection layer (He et al., 2016) and Batch Normalization layer (Ioffe & Szegedy, 2015) are consisted in Equation (4) and (5). The structure of GCN is motivated by Bresson & Laurent (2018).

The last layer edge embedding $e_{i,j}^L$ is then leveraged to compute the probabilities of different items being packed together, which can also be treated as a heat-map over the connectivity matrix of different items. The heat-map output is widely used to solve Traveling Salesman Problem (TSP) (Fu et al., 2021; Joshi et al., 2019), one of the most well-known CO problems. The packing probability $p_{i,j}$ of item $i$ and item $j$ can be calculated as follows:

$$p_{i,j} = \frac{exp(ReLU(W_6 e_{i,j}^L))}{exp(ReLU(W_6 e_{i,j}^L)) + 1} \qquad (6)$$

where $W_6 \in \mathbb{R}^{H \times H}$ and $p_{i,j} \in [0, 1]$.

### 4.2.2 NETWORK TRAINING

The network is trained via supervised learning and the ground-truth label can be generated through cutting stock (Gilmore & Gomory, 1961), which will be discussed in detail in Appendix D.1. The ground-truth labels can then be converted into a connection matrix where each element $\hat{p}_{i,j}$ denoted whether item $i$ and item $j$ are packed into one container, 1 for true and 0 for false, and $p_{i,j}, (i, j) \in E$ is the output of GCN.

In order to better get cluster characteristics of the heat-map, the loss should contain both the prediction errors and cluster biases. Specifically, the loss function includes two parts, the weighted cross-entropy loss $L_{ce}$ and the modular loss $L_m$.

$$L_{ce} = -\frac{1}{\gamma |V|} \sum_{i,j} \hat{p}_{i,j} \log p_{i,j} w_1 + (1 - \hat{p}_{i,j}) \log(1 - p_{i,j}) w_0 \qquad (7)$$

$$L_m = -\frac{1}{\gamma |V|} \sum_{i,j} p_{i,j} \cdot \hat{p}_{i,j} - p_{i,j} \qquad (8)$$

$L_{ce}$ tries to calculate the distance between the predicted probability and the ground-truth value. To avoid the positive and negative sample imbalance, we weighted $L_{ce}$ where $w_0 = \frac{N^2}{N^2 - 2N}$ and $w_1 = \frac{N^2}{2N}$. $w_0$ and $w_1$ are balanced coefficients relative to the edges belonging or not belonging to the ground-truth (Liu & Zhou, 2007). Minimizing $L_m$ means maximizing the probabilities that should be packed together indicated by ground-truth, which leverages *modularity* in Louvain (Blondel et al., 2008). Modularity measures the degree to which items in the same cluster are more densely connected than they are to items in other clusters, which is consistent with the objective of packing problems. The overall loss is defined as $L_{tot} = L_{ce} + \lambda L_m$ where $\lambda$ is a coefficient that balances the two losses.

### 4.3 DECODER

The heat-map generated by GCN encapsulates valuable cluster information on the items in the CCBPP. It provides probabilistic indications of item associations, with higher probabilities reflecting stronger correlations. Motivated by the First Fit (FF), the well-known greedy algorithm to solve 1DBPP, we first convert the heat-map into a permutation that determines the packing sequence, then

the FF algorithm is utilized to generate the final packing solution. Decoding the heat-map into a sequence also makes it possible to optimize the solution via reinforcement learning (RL), which will be discussed in Section 4.3.2.

The whole decode procedure is illustrated in Fig. 2b. In the beginning, several sequences are sampled from the generated heat-map by the Cluster Decode algorithm, which is highly parallelized. Following this decoding phase, the bin packing results are subsequently derived from the packing sequences via FF. The best result will be chosen and the network will be updated by backpropagation of the selected solution, hence the generated heat-map will be changed. After several rounds of network updates, the quality of the packing solution will be significantly improved.

### 4.3.1 CLUSTER DECODE

Since we obtain the packing probabilities according to the heat-map, decoding strategies can be leveraged to generate a sequence from the heat-map matrix, and the final solution is then obtained by FF. While greedy decoding and sample decoding, commonly employed in TSP, typically focus on the last item and overlook other items, packing problems require a comprehensive consideration of all packed items and the aggregation of relevant information. Cluster algorithms such as *K-Nearest Neighbors* can be leveraged to divide the items into several parts, but the divided parts of items may not be able to be packed in one bin, thus the final solution may be very different from $K$.

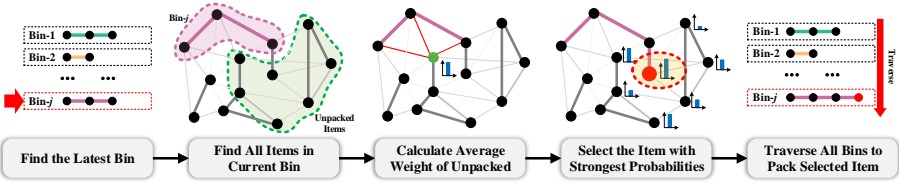

Figure 3: Procedure of the Cluster Decode

To address the packing constraints more effectively, the Cluster Decode algorithm is proposed to make full use of the generated heat-map and the relevant packing status. The algorithm ranks the items one by one, with each item's selection being determined by the maximum average connectivity probabilities with the items already placed in the latest bin, as depicted in Fig. 3. In this way, the aggregation information in the heat-map is better represented in the form of a sequence, so that the constraints will be considered during the decoding process while the information of the packing bins is also considered. The detailed Cluster Decode algorithm is described in Appendix C. At the beginning, one item is picked at random and the index is set as the first of the sequence, the item is put into the $j$th bin via FF. Then all bins are traversed and the latest open one $B_m$ is found. Since the newly selected item has the greatest possibility of being packed in the latest bin, the next item is picked by the average connectivity probabilities of all the items in $B_m$ represented by heat-map $P$. The sequence is generated until all $N$ items are visited.

### 4.3.2 ACTIVE SEARCH

It should be noted that while Cluster Decode provides both the packing sequence and the corresponding packing solution, there exists a notable discrepancy between the obtained solution and the optimal solution since the distribution of the training samples and the test samples may not be identical. In order to get more search guidance and react towards the solutions decoded by the Cluster Decode, reinforcement learning can be leveraged to fine-tune the packing sequence according to the solutions seen so far. Compared with supervised learning algorithms, reinforcement learning algorithms are guided by environmental reward signal so that it allows near-optimal solutions to be found without expert guidelines.

Motivated by Yu et al. (2016) and Hottung et al. (2022), we exploit RL-based Active Search to fine-tune the pre-trained network. Since only a few samples are leveraged for fine-tuning, we froze most of the graph network and update only one layer embedding with a RL loss function based on Policy Gradient (Williams, 2004). For each instance, solution $\pi$ can be repeatedly sampled by Cluster

Decode whose cost is $C(\pi) = -R(\pi)$, the opposite number of the occupied bins. The no-fixed parameters $w$ are adjusted to minimize $L(w)$:

$$L(w) = E_\pi[(C(\pi) - b)\nabla \log p_\theta(\pi|w)]$$

where $b$ denotes the average reward of the batch data and $p_\theta(\pi|w) = \prod_{t=1}^{N} p_\theta(a_t|s_t, w)$ (Kwon et al., 2020) , $p_\theta$ here is the probability of of generating a packing sequence based on the heat-map according to the Cluster Decode strategy mentioned above. Note that only the parameters $w$ are instance specific, while all other model parameters are identical for all instances.

## 5 PERFORMANCE EVALUATION

In this section, we first report the performance of our proposed Encoder-Decoder Model in solving two kinds of general CCBPP, single-class constrained BPP and multi-class constrained BPP as described in Section 3. Then we will conduct a case study of one typical application of general CCBPP, the Manufacturing Order Consolidation Problem. Finally, we conduct ablation studies to verify the effectiveness of the Encoder, Cluster Decode algorithm, and the effect of Active Search iterations.

We compare our algorithm with different heuristic algorithms and learning-based algorithms. The baseline heuristic methods include Random packing, First Fit Decreasing (FFD), which is one of the most well-known greedy algorithms to solve 1DBPP, and two typical population-based algorithms, Genetic Algorithm (GA) and Ant Colony Optimization algorithm (ACO) implemented by *scikit-opt*[1], whose modeling details are included in Appendix D.3.1. As for learning-based methods, since there is no learning-based related work about CCBPP, we re-implement Pointer Network (Vinyals et al., 2015) for CO problems with Policy Gradient (Bello et al., 2017) (PointNet in the following tables) as a baseline. The Greedy and Sampling are strategies to obtain a solution after Cluster Decode. Greedy strategy means we select the best solution after Cluster Decode, and sampling means we sample 32 solutions during the Cluster Decode and report the best. GCN-Cluster in the tables below means the Encoder-Decoder Model in this paper, Active Search (AS) is also compared with the Greedy and Sampling strategies. Note that exact solvers like OR-tools (Perron & Furnon, 2022) and gurobi (Bixby, 2007) cannot obtain the solution within a limited time. For example, it costs several hours to obtain one solution of single-class constrained BPP when $Q = 10, C = 3$, and besides, it is difficult to reproduce the exact algorithms provided in Borges et al. (2020) and da Silva & Schouery (2023), so we omit these results. The experiments are conducted on a Linux server with GeForce_RTX_3090 GPU and AMD EPYC 7542 32-Core Processor CPU@2.9GHz.

### 5.1 GENERAL CLASS-CONSTRAINED BIN PACKING PROBLEM

For general CCBPP, we conduct numerical experiments with simulated data. Since there are no known benchmarks for the CCBPP available in the literature, the datasets are generated following the convention of Borges et al. (2020) and da Silva & Schouery (2023). The sizes of items are random **integers** ranging from 10 to 25 while the bin capacity $B$ is 100. In single-class version, the number of different classes available $Q \in \{10, 20, 30\}$ and the class constraint value $C \in \{3, 5\}$. In the multi-class version, the class constraint value $C$ is 5, the number of different classes available $Q \in \{10, 20\}$ and each item belongs to $M$ classes, $M \in \{2, 3\}$. 6400 training instances and 200 test instances are generated via the cutting stock algorithm. For all the training and testing instances of general CCBPP, the optimal value is 20. Details of data generation are described in Appendix D.1.

Table 1 and Table 2 shows the performance evaluation of single-class constrained BPP. It can be shown that for all the settings, our algorithm outperforms all other baselines. FFD is not suitable for CCBPP and it performs even worse than Random Packing in most cases. Heuristic methods GA and ACO also obtain competitive solutions, but the time cost is higher. Note that the *scikit-opt* we use here is implemented with parallelization and caching techniques, so it runs more efficiently. Point Networks have achieved competitive performance on many routing problems (Bello et al., 2017), they do not perform well on our problems probably because the sequential decision model does not catch the relationship between items. Compared with Greedy and Sampling strategies, Active Search with RL further improves the quality of the solution by fine-tuning the network according to the characteristics of each test instance, meanwhile, the running time is also getting longer.

---

[1]https://github.com/guofei9987/scikit-opt

Table 1: Results of single-class constrained BPP when $C = 3$, and $Q \in \{10, 20, 30\}$, *Time(s)* is the average time to solve a single instance and the unit is second, *Bins* is the average cost of total bins, *Gap* refers to the gap between the optimal solution.

| Method | $Q = 10$ | | | $Q = 20$ | | | $Q = 30$ | | |
|---|---|---|---|---|---|---|---|---|---|
| | Bins | Gap | Time(s) | Bins | Gap | Time(s) | Bins | Gap | Time(s) |
| Random | 21.395 | 6.98% | 0.0095 | 22.04 | 10.20% | 0.0117 | 22.365 | 11.83% | 0.0106 |
| FFD | 21.58 | 7.90% | 0.0109 | 22.375 | 11.88% | 0.0109 | 23.09 | 15.45% | 0.0126 |
| GA | 20.99 | 4.95% | 14.51 | 21.00 | 5.00% | 16.43 | 21.00 | 5.00% | 17.86 |
| ACO | 20.81 | 4.05% | 92.00 | 20.935 | 4.68% | 93.89 | 21.00 | 5.00% | 96.85 |
| PointNet-Greedy | 21.41 | 7.05% | 0.0181 | 21.88 | 9.40% | 0.0181 | 22.11 | 10.55% | 0.019 |
| PointNet-Sampling | 21.07 | 5.35% | 0.40 | 21.03 | 5.15% | 0.41 | 21.06 | 5.30% | 0.67 |
| GCN-Cluster, Greedy | 21.00 | 5.00% | 0.045 | 21.03 | 5.15% | 0.051 | 21.29 | 6.45% | 0.056 |
| GCN-Cluster, Sampling | 20.965 | 4.83% | 0.40 | 21.00 | 5.00% | 0.41 | 21.00 | 5.00% | 0.62 |
| GCN-Cluster, AS(*Ours*) | **20.74** | **3.70%** | 10.14 | **20.91** | **4.55%** | 10.52 | **20.98** | **4.90%** | 14.05 |

Table 2: Results of single-class constrained BPP when $C = 5$ and $Q \in \{10, 20, 30\}$

| Method | $Q = 10$ | | | $Q = 20$ | | | $Q = 30$ | | |
|---|---|---|---|---|---|---|---|---|---|
| | Bins | Gap | Time(s) | Bins | Gap | Time(s) | Bins | Gap | Time(s) |
| Random | 21.005 | 5.03% | 0.0095 | 21.02 | 5.10% | 0.0120 | 21.03 | 5.15% | 0.0103 |
| FFD | 21.00 | 5.00% | 0.0118 | 21.08 | 5.40% | 0.0098 | 21.405 | 7.03% | 0.0109 |
| GA | 20.63 | 3.15% | 13.96 | 20.86 | 4.30% | 14.47 | 20.935 | 4.68% | 15.42 |
| ACO | 20.785 | 3.93% | 93.11 | 20.85 | 4.25% | 105.08 | 20.83 | 4.15% | 93.82 |
| PointNet-Greedy | 21.07 | 5.35% | 0.0156 | 21.00 | 5.00% | 0.023 | 21.21 | 6.05% | 0.0156 |
| PointNet-Sampling | 20.99 | 4.95% | 0.33 | 21.00 | 5.00% | 0.7609 | 21.00 | 5.00% | 0.667 |
| GCN-Cluster, Greedy | 21.00 | 5.00% | 0.036 | 21.00 | 5.00% | 0.055 | 21.00 | 5.00% | 0.0052 |
| GCN-Cluster, Sampling | 20.945 | 4.73% | 0.23 | 20.99 | 4.95% | 0.73 | 20.99 | 4.95% | 0.44 |
| GCN-Cluster, AS(*Ours*) | **20.47** | **2.35%** | 10.14 | **20.77** | **3.85%** | 10.52 | **20.82** | **4.10%** | 14.05 |

Table 3 shows the results of multi-class constrained BPP, our algorithm outperforms all other baselines for all the settings, with a more significant leading than that of single-class constrained BPP. Compared to the results of single-class constrained BPP in Table 2, the gap between all results and the optimal is larger and gradually increases as $M$ becomes larger. The results illustrate the difficulty of solving the multi-class version of CCBPP, especially when $M$ gets larger. From the results listed above, it can be inferred that our model has more prominent results when the input problem has more complex constraints such as multi-class constrained BPP.

Table 3: Results of multi-class constrained BPP when $C = 5$, $Q \in \{10, 20\}$ and $M \in \{2, 3\}$

| Method | $Q = 10, C = 5, M = 2$ | | | $Q = 10, C = 5, M = 3$ | | | $Q = 20, C = 5, M = 2$ | | | $Q = 20, C = 5, M = 3$ | | |
|---|---|---|---|---|---|---|---|---|---|---|---|---|
| | Bins | Gap | Time(s) | Bins | Gap | Time(s) | Bins | Gap | Time(s) | Bins | Gap | Time(s) |
| Random | 21.805 | 9.03% | 0.0169 | 25.33 | 26.65% | 0.0161 | 25.82 | 29.10% | 0.0167 | 43.36 | 116.80% | 0.0255 |
| FFD | 22.105 | 10.53% | 0.022 | 26.50 | 32.50% | 0.0256 | 27.67 | 38.35% | 0.026 | 44.26 | 121.30% | 0.0354 |
| GA | 20.995 | 4.98% | 52.29 | 21.785 | 8.93% | 68.90 | 21.955 | 9.78% | 73.76 | 27.875 | 39.38% | 56.74 |
| ACO | 20.97 | 4.85% | 106.83 | 21.78 | 8.90% | 131.36 | 21.93 | 9.65% | 112.22 | 26.83 | 34.15% | 119.64 |
| PointNet-Greedy | 21.88 | 9.40% | 0.0053 | 23.595 | 17.98% | 0.072 | 25.11 | 25.55% | 0.029 | 41.26 | 106.30% | 0.044 |
| PointNet-Sampling | 21.00 | 5.00% | 0.53 | 23.36 | 16.80% | 0.7208 | 23.70 | 18.50% | 0.8537 | 37.48 | 87.40% | 1.46 |
| GCN-Cluster, Greedy | 22.215 | 11.08% | 0.0527 | 23.595 | 17.98% | 0.0595 | 23.22 | 16.10% | 0.0616 | 32.095 | 60.48% | 0.0759 |
| GCN-Cluster, Sampling | 20.99 | 4.95% | 0.462 | 22.275 | 11.38% | 0.640 | 22.095 | 10.48% | 0.726 | 28.38 | 41.90% | 1.11 |
| GCN-Cluster, AS(*Ours*) | **20.91** | **4.55%** | 40.04 | **21.62** | **8.10%** | 38.77 | **21.515** | **7.58%** | 41.6 | **25.915** | **29.58%** | 58.21 |

## 5.2 CASE STUDY: ORDER CONSOLIDATION PROBLEM

For the Order Consolidation Problem, our datasets consists of the synthetic dataset with ground-truth and real supply chain dataset in September 2022. The real-world data contains orders in 24 days of September and orders ranging in size from 123 to 178 each day. The synthetic dataset is generated via cutting stock algorithms (Gilmore & Gomory, 1961) based on real-world data, which means the items in a sequence are created by "cutting" the bin so that the sequence can be perfectly packed and restored to the bin. Since the space utilization is 100%, the packing result can be treated as the ground-truth. In this experiment, we extract real orders to fulfill the feeder to get perfect space utilization, and the number of bins is set random from 3 to 14 to make the order numbers similar to the real ones. We generate orders of 2000 days for training and orders of 200 days for testing, the average ground-truth of the test labels is 9.11.

The results of synthetic data and real-world data are listed in Table 4. For two kinds of data, only one GCN is trained with 2000 synthetic data by cutting stock algorithms, and the trained network is employed on these two datasets, test synthetic datasets with 200 instances and real-world datasets with 24 instances. Since we already know the ground-truth of the synthetic data, the gap between different methods and the optimal solution can be obtained, whereas the gap of real-world data is omitted because the true labels cannot be calculated.

Table 4: Results of OCP on different datasets

| Method | Synthetic data | | | Real-world data | | |
|---|---|---|---|---|---|---|
| | Bins | Gap | Time(s) | Bins | Gap | Time(s) |
| Random | 11.705 | 28.48% | 0.038 | 17.68 | / | 0.050 |
| FFD | 11.865 | 30.24% | 0.041 | 17.79 | / | 0.052 |
| GA | 10.93 | 20.00% | 98.01 | 17.00 | / | 180.02 |
| ACO | 9.76 | 7.13% | 113.16 | 15.33 | / | 233.42 |
| PointNet-Greedy | 12.53 | 37.54% | 0.05 | 18.88 | / | 0.42 |
| PointNet-Sampling | 11.66 | 28.00% | 1.5 | 17.62 | / | 12.5 |
| GCN-Cluster, Greedy | 10.08 | 10.65% | 0.043 | 14.92 | / | 0.078 |
| GCN-Cluster, Sampling | 9.610 | 5.49% | 0.60 | 14.46 | / | 0.52 |
| GCN-Cluster, AS (*Ours*) | **9.325** | **2.36**% | 21.6 | **14.29** | / | 21.0 |

Results from Table 4 show that our algorithm outperforms all baselines. Although the running time of Random and FFD is short, the solution quality does not show any advantages. The time cost of heuristic methods GA and ACO is too high and the results are also not competitive. GCN can exhibit a high degree of clustering so it has shown good performance in our experiments. Active Search with RL further improves the quality of the solution by fine-tuning the network according to the characteristics of each test instance, meanwhile, the running time is also getting longer. Our algorithm can get an average of 9.325 bins with a gap of the best solution of 2.36% on synthetic data and an average of 14.29 bins on real-world data, respectively. Compared with the results on general CCBPP, our model also shows significantly promising performance in practical applications.

## 5.3 ABLATION STUDIES

We evaluate the effects of different parts of our Encoder-Decoder Model, and the results are as follows:

- Appendix E.1.1 The heat-map generated by the GCN Encoder provides more information than the others for the final result.
- Appendix E.1.2 The Cluster Decode algorithm does have a very positive impact on the performance of the whole framework.
- Appendix E.1.3 The learning curve of Active Search decreases more and more slowly when the number of iterations increases. A compromise may need to be made between solution quality and running time when the algorithm is applied in practice.

## 6 CONCLUSIONS

In this paper, we introduce and formulate a special variant of complex vector BPP, the Class-Constrained Bin Packing Problem, which has not been extensively studied in existing operational research. In particular, we propose a learning-based **Encoder-Decoder Model** to solve the BPP with multiple properties and more complex constraints. The GCN Encoder generates a heat-map that shows the probabilities of items packed together, then the Cluster Decode is leveraged to sequential decode the heat-map and fine-tune the solution according to the characteristics of the test sample with Active Search. Extensive experiments demonstrate that our methodology obtains high-quality solutions with a very small gap between the optimal while performing better than existing approaches on various settings of general CCBPP. Additionally, experiments also demonstrate that our Encoder-Decoder Model shows promising performance on OCP, one practical application of CCBPP. For future research, we are interested in trying to improve the generalities of our algorithms to different sizes and distributions of CCBPP. Applying our Encoder-Deocoder Model to other applications of CCBPP is also an interesting direction.

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

## A    MORE INTRODUCTION OF CCBPP

### A.1    FROM ACADEMIC VIEW

Bin packing problem (BPP) can be divided into two main categories, geometric BPP (2D and 3D BPP), and vector BPP (Christensen et al., 2017). The difference between these two problems is whether we should consider the geometrical constraints. In the vector BPP research field, there are plenty of works proposed to solve vector BPP with different constraints (Christensen et al., 2017; Epstein et al., 2010; Kellerer & Pferschy, 1999). Among them, Class-Constrained BPP is a typical variant. These research mainly focus on approximation heuristics (Shachnai & Tamir, 2001) or exact solutions (Borges et al., 2020). Unfortunately, learning-based methods have not been applied in this field so far.

### A.2    FROM APPLICATION VIEW

In general, BPP can be viewed as a resource allocation problem with resources from two kinds, incompatible resources and compatible resources (shared resources). The vanilla vector BPP only considers incompatible resources, i.e. all items are consuming accumulative resources. Actually, there are plenty of real-world applications corresponding to both kinds of resources. For example, an operation machine can only process 2 kinds of operation (compatible resource) with limited materials (incompatible resources) (Crévits et al., 2019). This is quite common in many real-world applications, we list some of these as follows.

- *Automatic Scaling in Cloud Computing* (Xiao et al., 2014). One server can hold several kinds of applications, each is consuming computing resources (CPU/Memory/Bandwidth).
- *Data-Placement Problem in Video-on-Demand* (Golubchik et al., 2000; Xavier & Miyazawa, 2006). The network loading is incompatible, but the data-file is shared among different online-users .
- *Production Planning* (Davis et al., 1993; Peeters & Degraeve, 2004). The raw material demand by each order is incompatible, while the production can be operated on the same machine from different orders.
- *Co-painting Problem* (Peeters & Degraeve, 2004). Item consuming the same color can share the same painting oil container while with different demand in amount.
- *Steel Mill Slab Problem* (Crévits et al., 2019). Different orders with the same production route can be operated on the same machine as long as their cumulative length does not exceed the slab-size.

These applications listed above are all typical Class-Constrained Bin Packing Problems. The optimization of these applications can bring numerous economic value by improving the efficiency.

# B MILP FORMULATIONS

## B.1 CCBPP

We present below an Integer Linear Programming formulation for the CCBPP as following Borges et al. (2020), in which we consider a 2D vector $x$ of size $N \times N$, a 2D vector $z$ of size $N \times Q$ and a vector $y$ of size $N$, all of the binary variables such that for all $i \in \{1, ..., N\}$ and $j \in \{1, ..., N\}$, $x_{ij}$ is a binary variable to indicate whether the $i$th item is packed in the $j$th container and $y_j$ is also binary to show if the $j$th container is occupied. $z_{jk}$ is 1 if class k is included in the $j$th container for $j \in \{1, ..., N\}$. For single-class constrained BPP, $z_{jk}$ is a one-hot vector, and for multi-class constrained BPP, there are $M$ ones in $z_{jk}$ for $j \in \{1, ..., N\}$.

$$minimize \; \sum_{j=1}^{N} y_j$$

$$subject \; to:$$

$$\sum_{i=1}^{N} s_i \cdot x_{ij} \leq B \;\; \forall \; j \in \{1, 2, 3..., N\}$$

$$\sum_{k=1}^{Q} z_{jk} \leq C \;\; \forall \; j \in \{1, 2, 3..., N\}$$

$$x_{ij} \leq z_{jc_i} \;\; \forall \; i, j \in \{1, 2, 3..., N\}$$

$$\sum_{j=1}^{N} x_{ij} = 1 \;\; \forall \; i \in \{1, 2, 3..., N\}$$

## B.2 OCP

The details of order consolidation are depicted in Fig. 4, A manufacturing order (item) involves multiple components (classes) supplied by a component feeder. The feeder has a fix-size thus it can only provide limited KINDS of components in one production process. Various orders can be combined into a new order (packing into a bin) to be produced, as long as the total kinds of components required for their production do not exceed the feeder's supply capacity. The primary objective of OCP is to minimize the total amount of orders after the consolidation process. The consolidation of orders will significantly reduce the equipment switching time, making it imperative to minimize the number of orders after consolidation to optimize operational efficiency.

Notably, the OCP presents heightened complexities compared to the general CCBPP due to the involvement of a substantial number of components and the potential inclusion of multiple components within a single order. This distinctive characteristic renders the attainment of high-quality solutions for the OCP more challenging in comparison to the general CCBPP.

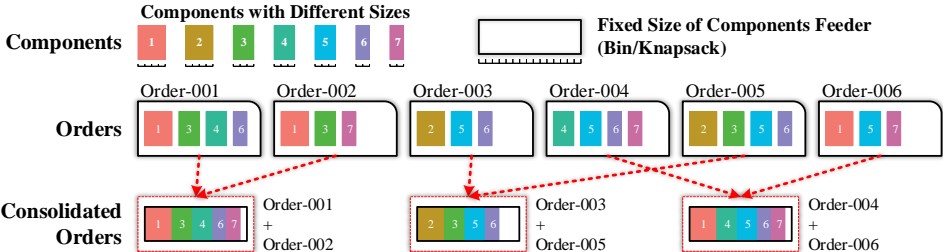

Figure 4: An illustration of OCP. Order-001 and 002, order-003 and 005, order-004 and 006 can be merged since the components collection after consolidation meet the constraints.

The whole process of OCP is illustrated in Fig. 5, which depicts the scenario of multiple orders $\{1, ..., N\}$ represented as printed circuit boards (PCBs) requiring the assembly of various components. The number of included components of all orders is $Q$. At the beginning of the production process, blank PCBs are placed onto the working space of a Surface Mount Technology (SMT) machine. Subsequently, a robotic arm picks up corresponding components from the feeder, placing them onto the PCBs. After finishing the current order, the SMT will be reset and switched for the next order. Notably, each kind of component $k$ occupies a specific "bandwidth" $s_k$ within the feeder. The feeder has a fixed size $C$ in bandwidth thus it can only provide limited kinds of components for one production process, while the supply of each kind of component can be ample in amount ($B = \infty$), as depicted in Fig. 5. Specifically, several orders can be combined into a new order to be produced, as long as the cumulative bandwidth of the component collections required for their production does not exceed the feeder's bandwidth limit $C$.

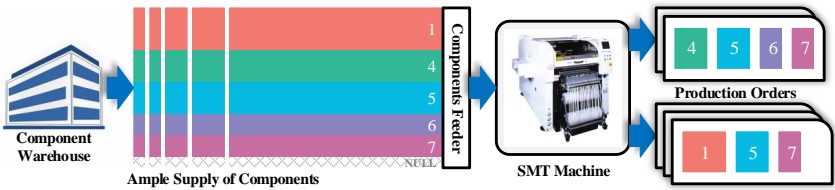

Figure 5: Feeders in PCB consolidation process

$$minimize \ \sum_{j=1}^{N} y_j$$

$$subject \ to:$$

$$\sum_{k=1}^{Q} z_{jk} \cdot s_k \leq C \ \ \forall \ j \in \{1, 2, 3..., N\}$$

$$x_{ij} \leq z_{jm} \ \ \forall \ i, j \in \{1, 2, 3..., N\}, m \in c_i$$

$$\sum_{j=1}^{N} x_{ij} = 1 \ \ \forall i \in \{1, 2, 3..., N\}$$

Compared with the formulation of CCBPP, the class one order belongs to is a subset, not one value of $\{1, 2..., Q\}$, which means one order may belong to multiple classes simultaneously. The limit of shared resource $B$ is equal to infinity and there are **only** class constraints in the problem.

## C  DETAILS OF CLUSTER DECODE

Here we list the algorithm details of Cluster Decode mentioned in Section 4.3.1.

---

**Algorithm 1** Cluster Decode

---

**Require:** The heat-map $P \in \mathbb{R}^{N \times N}$, the remaining items List set $\mathcal{L} = \{1, 2, ..., N\}$, $N$ empty bin
    set $B_1, B_2, ..., B_N = \emptyset$
**Ensure:** the packing items' sequence $S$
 1: Initialize items sequence $S = \emptyset$
 2: Randomly pop an $item_i$ from $\mathcal{L}$
 3: Add $item_i$'s index to the sorted list $S$
 4: **while** $\mathcal{L} \neq \emptyset$ **do**
 5:    Pack $item_i$ into $j$th bin according to FF algorithm
 6:    Add the index of $item_i$ into $B_j$
 7:    **for** $m = N...1$ **do**                      ▷ *find the latest open bin*
 8:        **if** $B_m \neq \emptyset$ **then**
 9:            break
10:        **end if**
11:    **end for**
12:    Calculate the average connectivity probabilities of items in $B_m$, $score = average(P\{k\}|k \in B_m)$
13:    Get the item index $i$ with the highest score $i = argmax(score\{k\}|k \in \mathcal{L})$
14:    Pop the $item_i$ with index $i$ from $\mathcal{L}$
15:    Add $item_i$'s index to the sorted list $S$
16: **end while**

---

# D    EXPERIMENT DETAILS

## D.1    DATA GENERATION

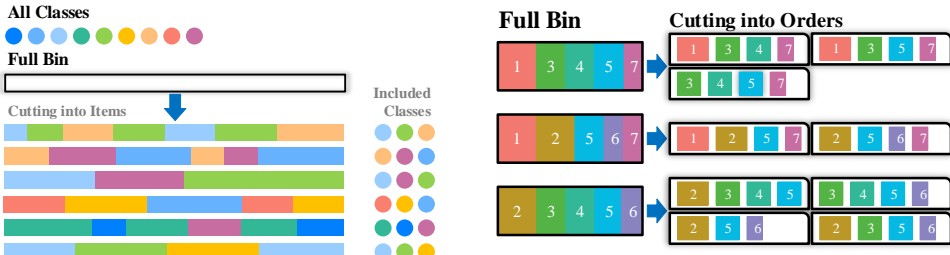

Figure 6: Ground-truth label generation of CCBPP and OCP by cutting stock algotithm.

It is difficult to obtain optimal solutions by exact solvers like OR-tools since it costs almost several hours to obtain one solution of single-class constrained BPP when $Q = 10, C = 3$ in our settings. As a result, we need to construct quantities of "perfect" solutions in other ways. Motivated by item sequences generation for online 3D Bin Packing Problems (Zhao et al., 2021b;a), our ground-truth labels can also be obtained by cutting stock (Gilmore & Gomory, 1961).

For general CCBPP, we first generate a sequence of valid items whose size threshold is $s_{min} \sim s_{max}$ following uniform distribution and the sum of which is equal to the bin size $B$, as shown in Fig. 6. Then we randomly choose $C$ classes over total $Q$ classes and assign the $C$ classes to the generated valid items. In this way, the items may be perfectly packed and restored to the bin, and the utilization is 100%. In all the single-class constrained BPP and multi-class constrained BPP settings, we generate 20 perfect sequences which means the optimal packing bin number of all the items is 20. The number of items in total 20 bins may vary in different instances, we choose the maximum number of items in all instances $N$ as the number constructed graph nodes and pad the other instances with 0. The pseudocode of our dataset construction algorithm is described in Algorithm 2.

For OCP, we generate a synthetic dataset according to the distributions of real-world datasets. The real-world data contains orders in 24 days of September and orders ranging in size from 123 to 178 each day. The components in each order contain a range from 955 to 1234, and there is a total of 2480 components included in all orders. The sizes of different components occupied are ranging from 1 to 6, and most of them are 1 or 2. The maximum size of the feeder is 282, as shown in Fig. 6.

The distribution of the real data may differ from the distribution of the generated data used to train the model. However, in practical applications, we can analyze the distribution of real data as much as possible and fit the most approximate dataset for training.

## D.2    EXPERIMENT HYPER-PARAMETERS

For the graph neural network, the number of GCN layers $L_{gcn}$ is 3, and the number of MLP layers $L_{mlp}$ is 3. The learning rate $lr$ is $5e-5$ with $1e-5$ weight decay, the 0 in edge features are replaced by $\epsilon = 0.3$, and the loss balance coefficient $\lambda = 0.3$ . Our network is trained for 20 epochs. It costs 2 hours to train the single-class constrained BPP and 3 hours to train the multi-class constrained BPP model.

For the Active Search, we update the parameters of the MLP's last layer, while keeping other parameters frozen. For sampling, 32 solutions per instance are sampled from the Cluster Decode procedure and report the best. The learning rate $lr$ is $5e-3$ with $1e-4$ weight decay, and for each instance, the search epoch is set as 30 for single-class constrained BPP experiments, and 50 for multi-class constrained BPP and OCP experiments.

---

**Algorithm 2** Dataset Construction

---

**Require:** Valid item size threshold $s_{min} \sim s_{max}$, bin capacity $B$, number of classes one bin can hold $C$, total number of classes $Q$, number of cutting bins $K$.
 1: **function** CONSTRUCTION OF ITEMS COLLECTION
 2:     **for** $j = 1...K$ **do**
 3:         initialize the used space of one bin $S_{used} = 0$, valid item list $\mathcal{L}_{valid} = \emptyset$
 4:         initialize class set $\mathcal{C}_{valid}$ which contains $C$ classes out of the total number $Q$
 5:         **while** $S_{used} \leq B$ **do**
 6:             generate one item where the size $s_i$ satisfies $s_{min} \leq s_i \leq s_{max}$, and $c_i \in \mathcal{C}_{valid}$
 7:             **if** $S_{used} + s_i < B$ **then**
 8:                 $S_{used} + = s_i$
 9:                 add the item into $\mathcal{L}_{valid}$
10:             **else if** $S_{used} + s_i == B$ **then**
11:                 add the item into $\mathcal{L}_{valid}$
12:                 break
13:             **else**
14:                 $S_{used} = 0$ , $\mathcal{L}_{valid} = \emptyset$
15:             **end if**
16:         **end while**
17:     **end for**
18:     **return** $\mathcal{L}_{valid}$
19: **end function**

---

For the baseline heuristic methods, the parameters of GA are 50 populations in the initial and 300 iterations. The distance matrix of ACO is set as the inverse value of the cosine similarity matrix and the number of populations and iterations are set as 50 and 100, respectively.

The parameters of PointNet are set as follows: the number of encoding layers is 2, and the number of multi-head is 4. The learning rate $lr$ is $1e-4$ with $1e-4$ weight decay.

### D.3 DETAILS OF BENCHMARK ALGORITHMS

#### D.3.1 GENETIC ALGORITHM AND ANT COLONY OPTIMIZATION MODELING

Genetic Algorithm (GA) (Mitchell, 1996) is a computational model searching for optimal solutions by simulating the natural evolution process with natural selection and genetic mechanism of Darwinian biological evolution theory. Genetic algorithms take all the individuals in a kind of population and use randomization techniques to guide an efficient search of an encoded parameter space. Among them, selection, crossover, and mutation constitute the genetic operation of the genetic algorithm. The parameter encoding, setting of the initial population, design of fitness function, design of genetic operation, and setting of control parameter, form the core of the genetic algorithm.

In computer science and operations research, the ant colony optimization algorithm (ACO) (López-Ibáñez, 2010) is a probabilistic technique for solving computational problems that can be reduced to finding good paths through graphs. Artificial ants stand for multi-agent methods inspired by the behavior of real ants. The pheromone-based communication of biological ants is often the predominant paradigm used. Combinations of artificial ants and local search algorithms have become a method of choice for numerous optimization tasks involving some sort of graph, e.g., vehicle routing and internet routing.

The output of both GA and ACO are sorted sequences. After the items' sequences are generated, different packing schemes such as First Fit (FF), Best Fit (BF), and Next Fit (NF) can be leveraged to pack the items in specific order so that the packing solution can be obtained. The input and output of GA are both full permutations of items to be packed and sizes and class information are needed to match the permutations to the final result. Specifically, $M$ full alignments as an initial population are randomly generated, and the maximum number of evolutionary generations $T$ is set. Then we need to set the adaptivity function, whose input is the sequence and information of items and output is the packing solution. Finally, the loop ends when the genetic evolution repeats $T$ rounds or reaches the iteration condition.

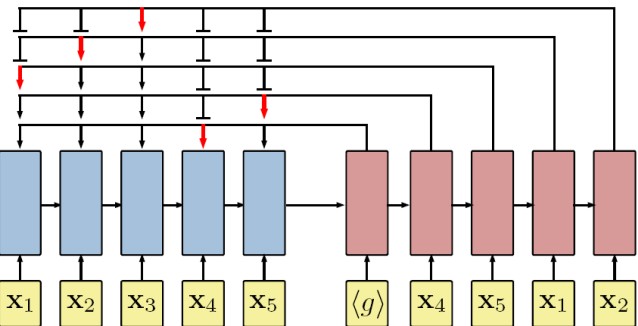

Figure 7: A pointer network architecture introduced by Vinyals et al. (2015)

For the ACO algorithm, the inputs include information about the items and also the distance matrix, which indicates the likelihood of choosing a path, the shorter the path the more likely it is that two items will be packed together. Here the distance matrix we set as the opposite number between the cosine similarity of each item class, i.e. the more similar the items are to each other and the smaller the distance between them, the more likely they are to be packed together.

### D.3.2 POINTNET MODELING

The Pointer Network is proposed to leverage the attention mechanism to solve the sorting problem of an input sequence. The traditional seq2seq model is not able to solve the problem that the vocabulary of the output sequence changes with the length of the input sequence. In some tasks, the input is strictly dependent on the input, or the output can only be selected from the input. In this case, the traditional seq2seq model ignores the priori information that the input can only be selected from the output. Pointer Networks are proposed to solve this problem. The relationship between the output elements and the input elements is calculated by the attention mechanism. The element with the largest attention value will be set as the output element. Actually, each output element is similar to a pointer to point to the input element. Note that each input element can only be pointed to one time to avoid the repetition in output sequence.

PointNet is widely used in routing problems where the output can directly be treated as one permutation of the input nodes. In the packing problem, since one packing solution corresponds to varies output sequences, supervised learning is inappropriate for solving Class-Constrained BPP. Reinforcement Learning provides an appropriate paradigm for training neural networks for combinatorial optimization, especially because these problems have relatively simple reward mechanisms that could be even used at test time (Bello et al., 2017). The input of the Pointer Network is $N$ items along with the size and class information, and the output is one specific permutation of all items. The packing solution can finally be obtained with the aid of FFD.

# E  MORE EXPERIMENT RESULTS

## E.1  ABLATION STUDIES

### E.1.1  BENEFITS OF THE ENCODER

The visualization of the heat-map is shown in Fig. 8. Orange points in Fig. 8 are the items visualized by t-SNE (van der Maaten & Hinton, 2008), an embedding technique that is commonly used for the visualization of high-dimensional data in scatter plots, the grey lines of the right figure represent the true label, which means two items are packed together in the optimal solution. The grey lines on the left figure show the connection of items when the value of the heat-map is larger than 0.2. It can be seen that these two figures show great similarities, which means our heat-map contains crucial information on the true labels.

Link Strength of Heatmap          Link Strength of True labels

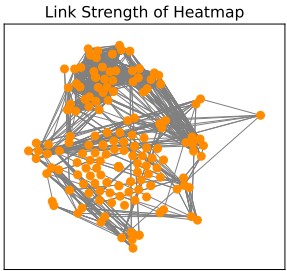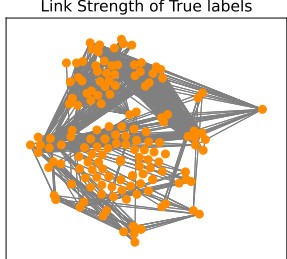

Figure 8: Visualization of heat-map

To verify the heat-map trained by GCN does have benefits to the final result, we compare different kinds of heat-map along with the Cluster Decode proposed in the Section 4.3 along with Active Search, and Table 5 and Table 6 list the results of CCBPP and OCP, respectively. Equal probability here means the elements in the initial heat-map are all equal to $\frac{1}{N}$ and $N$ is the number of items in one instance. Cos similarity in the following table is the *cosine similarity* of the class features of each item. PointNet (Vinyals et al., 2015) here is also leveraged an an Encoder to output a heat-map, the loss function is the same as ours. Results show the heat-map our GCN generated outperforms all the others on both CCBPP and OCP while the time overhead differs very little. GCN is best suited to capture the relationships of the items, and the heat-map generated by GCN obtains more information via the true labels, thereby contributing significantly to the final results.

Table 5: Comparisons on different heat-maps on CCBPP.

| Method | $Q = 10, C = 5$ | | | $Q = 20, C = 5$ | | |
|---|---|---|---|---|---|---|
| | Bins | Gap | Time(s) | Bins | Gap | Time(s) |
| Equal probs + Cluster Decode | 20.93 | 4.65% | 12.94 | 20.98 | 4.90% | 13.81 |
| Cos similarity + Cluster Decode | 20.95 | 4.75% | 12.37 | 20.98 | 4.90% | 12.94 |
| PointNet + Cluster Decode | 21.00 | 5.00% | 12.82 | 21.00 | 5.00% | 12.98 |
| GCN + Cluster Decode | **20.47** | **2.35%** | 10.98 | **20.77** | **3.85%** | 13.48 |

Table 6: Comparisons on different heat-maps on OCP.

| Method | Synthetic Data | | | Real-world Data | | |
|---|---|---|---|---|---|---|
| | Bins | Gap | Time(s) | Bins | Gap | Time(s) |
| Equal prob + Cluster Decode | 12.145 | 33.31% | 21.50 | 18.75 | / | 33.69 |
| Cos similarity + Cluster Decode | 12.135 | 33.20% | 21.39 | 18.67 | / | 35.77 |
| PointNet + Cluster Decode | 10.80 | 18.55% | 21.61 | 15.45 | / | 33.12 |
| GCN + Cluster Decode | **9.325** | **2.36%** | 21.26 | **14.29** | / | 21.88 |

### E.1.2  HOW CLUSTER DECODE ALGORITHM AFFECTS THE RESULT?

In this part, we want to figure out how the Cluster Decode algorithm proposed in Section 4.3.1 affects the result. Note that all decoding algorithms are combined with Active Search. Table 7 and

Table 8 list the CCBPP and OCP results of different heat-map decode schemes. Greedy Decode means the next item of one sequence has the maximum connectivity probability with the last item in the heat-map, whereas Sample Decode means the next item will be sampled according to the connectivity probability of the last item. The Greedy and Sample Decode schemes are different from the Greedy and Sampling strategies for obtaining solutions mentioned in Section 5. The parameters here are the same as mentioned in Appendix D.2. The result shows our Cluster Decode (Algorithm 1) achieves better solution quality than the Greedy Decode with similar time cost on synthetic data, which may due to the cluster information and packing state our decode scheme takes into account. As for real-world data, the results of Greedy Decode and Cluster Decode remain the same, which may be because the size of the test samples is too small. Sample Decode shows poor results in this part, which may be owing to the values in the heat-map do not differ much, and Sample Decode may bring noise to the final result. It can be proved from this experiment that the Cluster Decode algorithm does have a very positive impact on the performance of the whole framework.

Table 7: Comparisons of different decode schemes on CCBPP

| Method | $Q = 10, C = 5$ | | | $Q = 20, C = 5$ | | |
|---|---|---|---|---|---|---|
| | Bins | Gap | Time(s) | Bins | Gap | Time(s) |
| Greedy Decode | 20.51 | 2.55% | 10.16 | 20.94 | 4.70% | 13.14 |
| Sample Decode | 20.965 | 4.83% | 14.27 | 20.99 | 4.95% | 15.52 |
| Cluster Decode | **20.47** | **2.35%** | 10.14 | **20.77** | **3.85%** | 13.48 |

Table 8: Comparisons of different decode scheme on OCP

| Method | Synthetic Data | | | Real-world Data | | |
|---|---|---|---|---|---|---|
| | Bins | Gap | Time(s) | Bins | Gap | Time(s) |
| Greedy Decode | 9.450 | 3.73% | 21.9 | 14.29 | / | 19.5 |
| Sample Decode | 11.97 | 31.39% | 30.2 | 18.54 | / | 28.5 |
| Cluster Decode | **9.325** | **2.36%** | 21.6 | **14.29** | / | 21.0 |

### E.1.3 WHAT IS THE UPPER LIMIT OF THE FRAMEWORK?

In this section, we want to take OCP as an example and explore the impact of the number of AS iterations on the final result. It can be shown from Table 3 that the average results of synthetic data and real-world data are **9.325** and **14.29** when the search epoch is 50, how will the results change if the number of epochs is further increased?

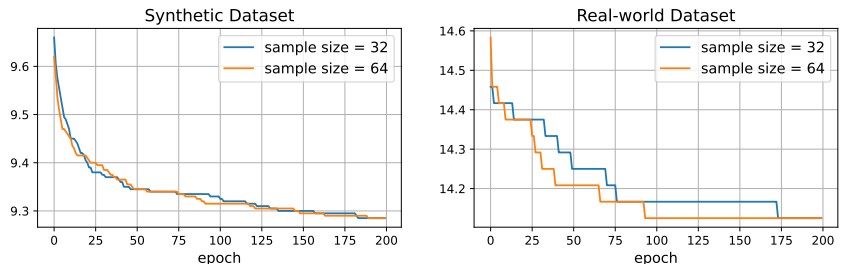

Figure 9: Learning curves of Active Search, with decode sample size = 32 and 64.

Fig. 9 shows the learning curve of the Active Search with RL, the sample size is set as 32 and 64 which means 32 or 64 solutions are sampled, and the best one is reported. Note that there are only 32 physical cores in the experimental server. The decline of final results becomes much slower as the number of iterations increases, whereas the running time always increases linearly. Results of the synthetic dataset and real-world dataset are 9.285 and 14.125 when $epoch = 200$, and the running time per instance is nearly 90s when the sample size is 32 and 180s when the sample size is 64, which is sometimes infeasible in real-world scenarios. A compromise may need to be made between solution quality and running time when the algorithm is applied in practice.

## E.2  1DBPP Results

1DBPP is a special case of CCBPP where $Q$ is 1 and $C$ is also 1. We want to figure out how the Encoder-Decoder Model works on 1DBPP. The setting of 1DBPP is the same as CCBPP where the capacity limit $B$ is 100. We consider two kinds of size settings here. For the small-scale items, the sizes range from 10 to 25, and for the large-scale items, the sizes range from 25 to 50. Note that the sizes of small-scale items and large-scale items are all integers in this experiment.

Table 9 shows the packing results of small items and large items. From the table, we can see our model can achieve the best results while spending less time than heuristic ones GA and ACO, and the gap to the optimal result is very small.

Comparing the results of different search strategies Greedy, Sampling, and AS, we can conclude that directly using the greedy algorithm can already get a promising result, which shows the representation ability of the heat-map for 1DBPP is remarkable. After adding AS, the resulting quality can still be significantly improved, which shows that the effect of the search is also very obvious for 1DBPP. In 1DBPP, only sizes of items are given and there are no class constraints so the relationship between different items is hard to get, but our GCN can still obtain the cluster information via the heat-map labels. The results of 1DBPP show the generalizability of our algorithm.

Table 9: Results of 1DBPP, for small items, the sizes are ranging from 10 to 25, and for large items, the sizes are ranging from 25 to 50. The optimal value of the bin is 20 for both datasets.

| Method | For small-scale items | | | For large-scale items | | |
| --- | --- | --- | --- | --- | --- | --- |
| | Bins | Gap | Time(s) | Bins | Gap | Time(s) |
| Random | 21.00 | 5.00% | 0.009 | 22.39 | 11.95% | 0.0075 |
| FFD | 21.00 | 5.00% | 0.011 | 22.17 | 10.85% | 0.0084 |
| GA | 20.59 | 2.95% | 8.26 | 21.00 | 5.00% | 7.02 |
| ACO | 20.77 | 3.85% | 20.77 | 21.00 | 5.00% | 75.99 |
| PointNet-Greedy | 21.00 | 5.00% | 0.011 | 22.34 | 11.70% | 0.0082 |
| PointNet-Sampling | 21.00 | 5.00% | 0.34 | 21.65 | 8.25% | 0.26 |
| GCN-Cluster, Greedy | 20.79 | 3.95% | 0.031 | 20.86 | 4.30% | 0.032 |
| GCN-Cluster, Sampling | 20.44 | 2.20% | 0.21 | 20.36 | 1.80% | 0.18 |
| GCN-Cluster, AS(*Ours*) | **20.24** | **1.20%** | 5.14 | **20.27** | **1.35%** | 4.13 |

## E.3  Generalization to different distributions

We investigate the generalization capability of our algorithm to the problem instances generated from different distributions. Since different $C$, $Q$, and $M$ mean different problems, we mainly explore the generalization of item sizes. We sample $s$ from normal distributions $N(\mu, \sigma^2)$ where $\mu$ and $\sigma$ are the expectation and the standard deviation. Two normal distributions $s \sim N(15, 5^2)$ and $s \sim N(20, 5^2)$ are adopted here. The test datasets of these two distributions are generated at random, not by cutting stock, so the optimal number is unknown. Another uniform distribution $s \sim U(25, 50)$ generated also by cutting stock is also tested here, and the optimal bins of the uniform distribution are 20.

Table 10 and Table 11 list the results of CCBPP when $C = 5, Q = 10$ and $C = 5, Q = 20$. The GCN model is trained on the uniform distribution $U(10, 25)$ without any fine-tuning. For uniform distribution $s \sim U(25, 50)$, our method does not show advantages with heuristic algorithms GA and ACO, it may be due to the sizes of larger uniform distribution do not have any overlap with the training distribution. For the other two normal distributions, ours performs better than others, but with a weak advantage. It may further improve the generalization capability if we better design the structure of our Graph Neural Networks, which we leave to future work.

## E.4  Results on large-scale benchmarks

We conducted experiments on $N = 300$ and $N = 500$, as shown in Table 12 and Table 13. The dataset is also generated following Appendix.D.1, and the optimal bins are 30 and 50 respectively. The results show that our proposed method still has advantages compared to other baselines. Larger-scale experiments when $N > 1000$ are not conducted at present due to the resource limitation.

Table 10: Results from different distributions when $C = 5$ and $Q = 10$.

| Method | $s \sim U(25, 50)$ | | | $s \sim N(20, 5^2)$ | | | $s \sim N(15, 5^2)$ | | |
|---|---|---|---|---|---|---|---|---|---|
| | Bins | Gap | Time(s) | Bins | Gap | Time(s) | Bins | Gap | Time(s) |
| Random | 22.44 | 12.20% | 0.0070 | 40.655 | / | 0.0152 | 30.125 | / | 0.012 |
| FFD | 22.15 | 10.75% | 0.0095 | 40.15 | / | 0.0179 | 30.12 | / | 0.016 |
| GA | 21.00 | 5.00% | 13.18 | 39.875 | / | 17.2 | 29.74 | / | 14.45 |
| ACO | 21.00 | 5.00% | 93.86 | 39.945 | / | 105.08 | 29.74 | / | 97.83 |
| PointNet-Greedy | 22.34 | 11.70% | 0.0098 | 40.63 | / | 0.023 | 30.07 | / | 0.019 |
| PointNet-Sampling | 21.68 | 8.40% | 0.33 | 40.23 | / | 0.7609 | 29.89 | / | 0.63 |
| GCN-Cluster, Greedy | 22.29 | 11.45% | 0.036 | 40.08 | / | 0.055 | 30.05 | / | 0.0052 |
| GCN-Cluster, Sampling | 21.91 | 9.55% | 0.23 | 39.81 | / | 0.73 | 29.67 | / | 0.60 |
| GCN-Cluster, AS(*Ours*) | **21.00** | **5.00%** | 10.14 | **39.74** | / | 10.52 | **29.59** | / | 14.05 |

Table 11: Results from different distributions when $C = 5$ and $Q = 20$.

| Method | $s \sim U(25, 50)$ | | | $s \sim N(20, 5^2)$ | | | $s \sim N(15, 5^2)$ | | |
|---|---|---|---|---|---|---|---|---|---|
| | Bins | Gap | Time(s) | Bins | Gap | Time(s) | Bins | Gap | Time(s) |
| Random | 22.405 | 12.03% | 0.0070 | 40.95 | / | 0.0152 | 30.33 | / | 0.0124 |
| FFD | 22.115 | 10.58% | 0.0164 | 40.60 | / | 0.018 | 30.53 | / | 0.016 |
| GA | 21.00 | 5.00% | 12.21 | 40.185 | / | 21.33 | 29.75 | / | 18.67 |
| ACO | 21.00 | 5.00% | 96.74 | 40.19 | / | 94.81 | 29.65 | / | 89.31 |
| PointNet-Greedy | 22.34 | 11.70% | 0.012 | 40.87 | / | 0.024 | 30.08 | / | 0.020 |
| PointNet-Sampling | 21.69 | 8.45% | 0.415 | 40.48 | / | 0.77 | 29.85 | / | 0.77 |
| GCN-Cluster, Greedy | 22.08 | 10.40% | 0.040 | 40.87 | / | 02024 | 30.08 | / | 0.020 |
| GCN-Cluster, Sampling | 21.68 | 8.40% | 0.29 | 40.15 | / | 0.72 | 29.84 | / | 0.62 |
| GCN-Cluster, AS(*Ours*) | **21.00** | **5.00%** | 12.02 | **39.85** | / | 15.91 | **29.60** | / | 16.33 |

There are some excellent works in the direction of combinatorial optimization that transplant models trained on small-scale problems to solve large-scale ones (Li et al., 2021; Fu et al., 2021; Son et al., 2023), which should be inspiring and helpful to solve large-scale CCBPP with lower time complexity for the future research.

Table 12: Results when $N = 300$.

| Method | $Q = 10, C = 5$ | | | $Q = 20, C = 5$ | | |
|---|---|---|---|---|---|---|
| | Bins | Gap | Time(s) | Bins | Gap | Time(s) |
| Random | 31.25 | 4.17% | 0.0136 | 31.07 | 3.57% | 0.0134 |
| FFD | 31.050 | 3.50% | 0.0141 | 31.125 | 3.75% | 0.0145 |
| GA | 30.58 | 1.93% | 27.99 | 30.62 | 2.07% | 29.753 |
| ACO | 30.625 | 2.08% | 163.87 | 30.595 | 1.98% | 179.704 |
| PointNet-Greedy | 31.09 | 3.63% | 0.034 | 31.19 | 3.97% | 0.0305 |
| PointNet-Sampling | 31.00 | 3.33% | 0.3757 | 31.00 | 3.33% | 0.5056 |
| GCN-Cluster, Greedy | 31.00 | 3.33% | 0.0644 | 30.995 | 3.32% | 0.0702 |
| GCN-Cluster, Sampling | 30.925 | 3.08% | 0.6250 | 30.93 | 3.10% | 0.6911 |
| GCN-Cluster, AS(*Ours*) | **30.545** | **1.82%** | 21.0235 | **30.57** | **1.90%** | 20.16 |

Table 13: Results when $N = 500$.

| Method | $Q = 10, C = 5$ | | | $Q = 20, C = 5$ | | |
|---|---|---|---|---|---|---|
| | Bins | Gap | Time(s) | Bins | Gap | Time(s) |
| Random | 51.86 | 3.72% | 0.0409 | 51.95 | 3.90% | 0.0409 |
| FFD | 51.4150 | 2.83% | 0.0308 | 51.76 | 3.52% | 0.0315 |
| GA | 50.90 | 1.80% | 72.65 | 51.00 | 2.00% | 76.65 |
| ACO | 50.97 | 1.94% | 388.10 | 51.00 | 2.00% | 418.45 |
| PointNet-Greedy | 51.91 | 3.82% | 0.0509 | 52.06 | 4.12% | 0.0509 |
| PointNet-Sampling | 51.22 | 2.44% | 0.7997 | 51.46 | 2.92% | 0.8793 |
| GCN-Cluster, Greedy | 51.4650 | 2.93% | 0.1300 | 51.6450 | 3.29% | 0.1388 |
| GCN-Cluster, Sampling | 51.0050 | 2.01% | 1.4969 | 51.02 | 2.04% | 1.5866 |
| GCN-Cluster, AS(*Ours*) | **50.84** | **1.68%** | 63.72 | **50.99** | **1.98%** | 64.52 |

## E.5 THE ENCODER-DECODER FOR OTHER PROBLEMS

Our proposed Encoder-Decoder Model also has the potential to generalize to other problems, we try to prove that on multi-dimensional knapsack problem (MDKP). The multi-dimensional knapsack problem is a classic optimization problem in which a set of items with multiple attributes or

dimensions (such as weight, volume, or value) must be selected to maximize a given objective while satisfying certain constraints. Since there is only one bin in MDKP and we only need to decide whether one item should be packed in or not, the heat-map in the model can be treated as a vector in this problem, and the Decoder is the same as the greedy strategies. The features of the vertices are the properties of the items, and the edge features are filled with 1.

In this section, we use the performance (the achieved value in a knapsack) by GLOP implemented in the OR-tools (Perron & Furnon, 2022) as a baseline. The value/weight greedy, Reinforcement Learning, and Learning-to-Rank are used as comparisons following Woo et al. (2022), gumbel trick is an efficient sequence sampling method also proposed in Woo et al. (2022).

Table 14: Results of MDKP on different datasets, $N$ and $k$ denote the number of items and the size of knapsack resource dimensions, respectively. $c$ denotes the correlation of weight and value of items. *Time* is the average time to solve a single instance and *Gap* represents the ratio to the GLOP algorithm.

| Method | $N = 50, k = 3, c = 0.0$ | | $N = 50, k = 3, c = 0.9$ | | $N = 100, k = 10, c = 0.0$ | | $N = 100, k = 10, c = 0.9$ | | $N = 200, k = 20, c = 0.0$ | | $N = 200, k = 20, c = 0.9$ | |
| --- | --- | --- | --- | --- | --- | --- | --- | --- | --- | --- | --- | --- |
| | Gap | Time(s) | Gap | Time(s) | Gap | Time(s) | Gap | Time(s) | Gap | Time(s)e | Gap | Time(s) |
| GLOP | 100% | 0.066 | 100% | 0.11 | 100% | 0.24 | 100% | 0.24 | 100% | 1.63 | 100.0% | 0.78 |
| Greedy | 98.16% | 0.00039 | 90.26% | 0.00052 | 100.86% | 0.00059 | 98.85% | 0.00076 | 102.57% | 0.001 | 101.72% | 0.00097 |
| RL | 90.30% | 2.32 | 99.55% | 0.48 | 93.97% | 1.85 | 101.26% | 0.96 | 97.27% | 3.02 | 102.61% | 3.68 |
| RL-Sampling | 82.51% | 2.02 | 99.19% | 0.50 | 89.83% | 1.73 | 101.30% | 0.98 | 94.87% | 2.94 | 102.65% | 3.58 |
| RD | 87.94% | 0.092 | 96.14% | 0.02 | 91.00% | 0.054 | 97.60% | 0.024 | 95.28% | 0.054 | 100.52% | 0.034 |
| RD-Gumbel | 90.69% | 0.10 | 99.66% | 0.041 | 93.49% | 0.069 | 101.15% | 0.054 | 95.93% | 0.081 | 101.36% | 0.093 |
| GCN-Greedy | 99.54% | 0.0015 | 97.34% | 0.0012 | 101.07% | 0.0016 | 100.23% | 0.0011 | 102.61% | 0.0034 | 102.20% | 0.0031 |
| GCN-Gumbel | 99.64% | 0.017 | 99.00% | 0.019 | 101.76% | 0.030 | 101.26% | 0.026 | 103.00% | 0.044 | 102.71% | 0.047 |
| GCN-AS | **101.59%** | 0.061 | **101.00%** | 0.051 | **104.30%** | 0.076 | **104.94%** | 0.072 | **103.81%** | 0.091 | **105.30%** | 0.083 |

Table 14 shows the results of datasets with different scales. GCN-Greedy, GCN-Gumbel, and GCN-AS provided in Table 14 are the results of training the network and introducing different search algorithms. As mentioned above, only one bin is included in MDKP, the heat-map we get here is only one-dimensional data, and the value represented in the data is the probability of this item being packed into the knapsack. According to this one-dimensional representation, some search algorithms can be combined to obtain the final result. From the results, we can see that the results of AS are greatly improved compared with Greedy and Gumbel Search, but the time cost is relatively large. Gumbel search can also improve the results by introducing a certain random mechanism, yet the improvement is not obvious. It can be concluded from Table 14 that our framework GCN-AS achieves the best results on all datasets, while the time cost is the lowest among all the learning-based methods.

As $N$ increases, all methods show longer inference time, but the increment gap of GLOP is much larger than other methods.Our method shows superior performance over GLOP as $N = 50$ with less time-consuming. The greedy algorithm shows a very good advantage in running time and solution quality, especially when $c = 0.0$. RL and RD achieve good performances when $c = 0.9$ but the time complexity of RL is very high. Our algorithm shows even better results than all other algorithms on both solution quality and running time compared to smaller $N$.

