# OpenReview forum: "Learning to solve Class-Constrained Bin Packing Problems via Encoder-Decoder Model"
_ICLR.cc/2024/Conference — ICLR 2024 poster_

### Official Review · Reviewer_hLWM · 2023-10-29

**Soundness:** 2 fair
**Presentation:** 2 fair
**Contribution:** 2 fair
**Rating:** 6
**Confidence:** 4

**Summary:**

The authors aim to solve a special 1D-BPP with class constraints. First, train a heatmap to represent the probability. Second, use a decode strategy to generate a solution from the heatmap while satisfying the constraints. In addition, it could be finetuned with RL.

I admit it is a new problem, but the action space is much smaller than traditional 3D-BPP.

**Strengths:**

The problem is new and the authors proposed a framework (which might be already used in TSP and VRP)  to solve the problem well.

The authors conduct comprehensive experiments.

The ablations study is provided.

**Weaknesses:**

The used heatmap method is not new and a similar method has been used in TSP and VRP.

The studied topic CCBPP and problem is less-studied and uncommon to see.

The method is not explained well and clearly in the main body such as cluster decode and active search.

The active search seems to be fine-tuning with RL. I do not see anything novel.

The paper is not easy to follow.

**Questions:**

"we use class cosine similarities as edge feature" How to calculate the edge feature?

What's the size number N set in the experiments?

What's the dataset used for finetuning in experiments?

Is it a fair comparison with the baseline method without the finetuning dataset?

Why did the active search improve so significantly?

Could you clearly explain how to select items? how do choose the current bin to place the selected items while satisfying constraints?

Could other constraints be considered rather than just class numbers?

The constraints seem not as complex as the constraints in 3D-BPP. In the seq2seq problem, it could be easily handled by the mask function.

---

> ### Author Response · Authors · 2023-11-18
> **Response to Reviewer hLWM (1/2)**
>
> **Dear Reviewer hLWM:**
>
> We appreciate your insightful thoughts of our work. We will address your concerns as follows:
>
> **Q1: The used heatmap method is not new and a similar method has been used in TSP and VRP.**
>
> We appreciate the nice works [1,2] which leverage heat-map to boost the performance of TSP. Our heat-map is different from those applied in routing algorithm from two aspects.
>
> * Firstly, our motivation to propose the heat-map is to solve complex vector bpp such as CCBPP. Learning-based packing problems often use sequential models to construct solutions and they can also get promising results on geometric BPPs. In this paper, we offer another perspective to solve complex vector BPP such as CCBPP. Concretely, we introduce the connection matrix as a label to represent whether different items are packed together in the optimal result, which can better represent the cluster information in the packing problems.
> * Secondly, the training details are slightly different from [1,2]. Our heat-map consider connection information including modularity information while conventional research focus on node information and geometrical information. Besides, the ground-truth data generation via cutting stock algorithm mentioned in Appendix D.1 cost much lower time than the TSP labels generation by concorde.
>
> [1] Joshi, Chaitanya K. et al. “An Efficient Graph Convolutional Network Technique for the Travelling Salesman Problem.” ArXiwo qianv abs/1906.01227 (2019): n. pag.
>
> [2] Fu, Zhang-Hua et al. “Generalize a Small Pre-trained Model to Arbitrarily Large TSP Instances.” ArXiv abs/2012.10658 (2020): n. pag.
>
> **Q2: The studied topic CCBPP and problem is less-studied and uncommon to see.**
>
> We are very sorry we don't claim the significance and generality of CCBPP clearly in our paper,  more about the CCBPP will be added into the revision of the manuscript. As far as we're concerned, the points can be listed as follows:
>
> 1. **In the academic view, CCBPP as well as vector BPP are fully studied beyond neural combinatorial optimization community.**
>
>     BPP can be divided into two main categories, geometric BPP (2D and 3D BPP), and vector BPP. The difference between these two problems is whether we should consider the geometrical constraints.
>     In the vector BPP research field, there are plenty of works proposed to solve vector BPP with different constraints. Among them, class-constrained BPP is a typical variants. These research mainly focus on approximation heuristics or exact solutions. Unfortunately, learning-based methods have not been applied in this field so far.
>
> 2. **In the application view, CCBPP is commonly encountered in BPP field with wide applications.**
>
>     In general, BPP can be viewed as a resource allocation problem with resources from two kinds, incompatible resources and compatible resources (shared resources). The vanilla vector BPP only considers incompatible resources. Actually, there are plenty of real-world applications corresponding to both kinds of resources. For example, an operation machine can only process 2 kinds of operation (compatible resource) with limited materials (incompatible resources) . This is quite common in many real-world applications, *Automatic Scaling in Cloud Computing*, *Data-Placement Problem in Video-on-Demand* , *Production Planning* , *Co-painting Problem* , *Steel Mill Slab Problem* are all typical real-world applications of CCBPP.
>
> **Q3: cluster decode and active search are not explained well, could you clearly explain how to select items?**
>
> We are very sorry details of Cluster Decoder are not explained clearly in our paper.
> The purpose of the Cluster Decoder is to map the trained heatmap to the final packing result. It contains two parts: Cluster Decode and Active search. The Cluster Decode strategy maps the heatmap to a packing sequence. The packing sequence can be transferred to the final packing result according to First Fit (FF) scheme which is commonly used in packing problem, as mentioned in section 4.3.1. For each selected item, FF will be placed to the smallest number of the bin if the constraints are satisfied.
>
> Active Search fine-tunes the last layer of GCN based on the corresponding packing results obtained by the packing sequence along with the FF. In this way, the generated heatmap can be more consistent with the characteristics of the test instance itself and can also decode better results.

---

> ### Author Response · Authors · 2023-11-18
> **Response to Reviewer hLWM (2/2)**
>
> **Q4: the active search is not novel, yet it boots the final results significantly.**
>
> Active search is not the main contribution of this paper, it is combined with the  cluster decode strategy so that the instance-wise solution can be obtained by only changing the parameters of the last layer of Encoder. In the ablation experiments E.1.1 and E.1.2 provided in the appendix, we proved that the heat-map generated by the Encoder and the cluster decode strategy are also effective. These two parts also play greater roles in improving experimental results.
>
> Active Search is an excellent search strategy that can fine-tune the network according to the characteristics of each example. For combinatorial optimization problems, it is common to have differences between training samples and test samples, so there are many learning methods that combine certain search strategies to further improve the quality of the solution[1-5].
>
> [1]	Fu, Zhang-Hua et al. “Generalize a Small Pre-trained Model to Arbitrarily Large TSP Instances.” ArXiv abs/2012.10658 (2020): n. pag.
>
> [2] Choo, Jinho et al. “Simulation-guided Beam Search for Neural Combinatorial Optimization.” Advances in neural information processing systems 35 (2022).
>
> [3] Kool, Wouter et al. “Deep Policy Dynamic Programming for Vehicle Routing Problems.”
> Integration of AI and OR Techniques in Constraint Programming (2021).
>
> [4] Qiu, Ruizhong et al. “DIMES: A Differentiable Meta Solver for Combinatorial Optimization Problems.” Advances in neural information processing systems 35 (2022).
>
> [5] Son, Jiwoo et al. “Meta-SAGE: Scale Meta-Learning Scheduled Adaptation with Guided Exploration for Mitigating Scale Shift on Combinatorial Optimization.” International Conference on Machine Learning (2023).
>
> **Q5: How to calculate the edge features?**
>
> The edge features are set as the cosine similarities corresponding to the one-hot form of class features, as mentioned in section 4.2.1.
>
> **Q6:  What's the size number N set in the experiments?**
>
> We are sorry we didn't make it clear in the text. When generating the ideal data set, we generated 20 bins with 100% utilization. Since the item size is 10~25 and the box capacity is 100, we can put a maximum of 10 items in each bin and a minimum of 4. In order to ensure the consistency of training, we set N = 20 * 10 = 200. If the number in one bin does not reach 10, we will add 0, as stated in Appendix D.1.
>
> **Q7: What is the dataset used for finetuning in experiments? Is it fair to compare to baselines without finetune dataset?**
>
> The finetune is performed on **instance level**. The solution of each instance is further improved based on active search, which is denoted as 'finetune' in our paper. In other words, this finetuning procedure is performed on each test instance *separatedly*, and the dataset for finetuning is the instance itself with iterated costs $C(\pi) = -R(\pi)$.
>
> On the other hand, since active search is not the main contribution of this paper, we compare the performance of our model without finetune to these baselines (See experiments E.1.1 and E.1.2 provided in the appendix). These experiments demonstrate that our Encoder-Decoder model without finetune (GCN-Cluster-Greedy and GCN-Cluster-Sampling) also outperforms baselines (PointNet-Greedy and PointNet-Sampling), which validates the advantages of our model.
>
>
> **Q8:  Could other constraints be considered rather than just class numbers?**
>
> The class-constrains in CCBPP changes the relationship between different items and it  can be reflected in problem formulation. If the mask is leveraged to judge the constraint in the form of a sequence, such as using seq2seq to construct the problem model, the results may not be so promising, as shown in our paper.
>
> If there are some **unstable** and **personalized features** in the actual scene, which can not be encoded by the network, adding restrictions can be added to Cluster Decoder with specific mask function so that both the connectivity probabilities denoted by the heatmap and the complex constrains can be considered in the final result.

---

> ### Author Response · Authors · 2023-11-21
> **Reminder of Reviewer's feedback**
>
> **Dear Reviewer hLWM :**
>
> We respectfully remind you that it has been more than 3 days since we submitted our rebuttal. We would appreciate your feedback on whether our response has addressed your concerns.
>
> In response to your comments, we have answered your concerns and improved the paper in the following aspects:
>
> * We claimed the novelty of  leveraging the heatmap from two aspects.
>
> * We provided more introduction about CCBPP both **from academic view** and **from application view**, and this part was added to the appendix of our updated manuscript.
>
> * We claimed the details of cluster decode and active search and this part will be modified in the revision of our manuscript.
>
> * We emphasized that the encoder-decoder structure also played important roles on improving experimental results according to the ablation study, and the active search is not our main contribution.
>
> * We responded to some details of our paper including the edge calculation, $N$ setting in our experiment and the dataset used for Active Search finetuning.
>
> * We explained that the extra complicated constraints could be added to the cluster decode part using specific mask function.
>
> Thanks again for your valuable review. We are looking forward to your response and are happy to answer any future questions.

---

### Official Review · Reviewer_Rj4P · 2023-10-31

**Soundness:** 3 good
**Presentation:** 3 good
**Contribution:** 3 good
**Rating:** 6
**Confidence:** 4

**Summary:**

This paper presents a neural network solver for the bin packing problem. The neural network model is built under the encode-decoder framework, where the encoder network is a graph neural network to modulate the problem, and the decoder module contains heuristics, neighborhood search, and RL-like active search. Experiments are conducted on

**Strengths:**

* The problem of bin packing problem and its variants are important and worth studying.
* The methodology presented in this paper seems technically sound.
* The authors conduct extensive experiments on different variants of the bin packing problem. The design of data generation with ground-truth labels in the order consolidation problem is interesting.
* This paper is well-written and easy to follow.

**Weaknesses:**

* The encoder-decoder pipeline is quite common in machine learning solvers for combinatorial optimization problems. The authors make some specific adaptations for the bin packing problem, while in general, the results are not too surprising under such a framework.
* It will be better to have more insights into the bin packing problem and what machine learning can help to inspire future researchers.

**Questions:**

* Can you plot the gap vs time for different methods?

---

> ### Author Response · Authors · 2023-11-18
> **Response to Reviewer Rj4P**
>
> **Dear Reviewer Rj4P:**
>
> We appreciate your positive recognition of our work，we will address your concerns as follows:
>
> **Q1: The novelty of our proposed Encoder-Decoder framework**
>
> We are very sorry we don't compare our encoder-decoder framework over other structures in our paper, the relative contents will be added to the revision of our manuscript. The encoder-decoder models for the combinatorial problems such as [1-3] are sequential models, which means the neural solvers are learned by the constructed sequences. In the context of CCBPP, an optimal packing result always corresponds to numerous packing sequences since shuffling the order of items within each bin does not affect the packing solution but will directly alter the packing sequence. As a result, sequential model may not be suitable.
>
> Overall, the novelty of our proposed encoder-decoder framework are listed as follows:
>
> * In the Encoder part, we introduce the connection matrix as a label to train the heat-map so that it can bring richer information than a packing sequence.
>
> * The Cluster-Decoder is designed to generate instance-wise packing solutions from heat-map, which successfully closes the gap between a heat-map and a final packing solution.
>
> * Our encoder and decoder are separated so that it is more flexible when applied in actual scene. The encoder is able to capture more problem-specific information, and the decoder can fine-tune the final solution at instance-wise level.
>
> [1] Vinyals, Oriol, Meire Fortunato, and Navdeep Jaitly. "Pointer networks." Advances in neural information processing systems 28 (2015).
>
> [2] Li, D., et al. "Solving Packing Problems by Conditional Query Learning." International Conference on Learning Representations (ICLR) Conference 2020.
>
> [3] Zhang, Jingwei, Bin Zi, and Xiaoyu Ge. "Attend2pack: Bin packing through deep reinforcement learning with attention." arXiv preprint arXiv:2107.04333 (2021).
>
> **Q2:  More insights into the bin packing problem**
>
> As far as we are concerned, our contributions to other BPPs can be listed as follows:
>
> * Conventional learning-based COP algorithms mainly focus on solving geometric BPP like 3DBPP, In this paper, we bring another complex vector BPP, CCBPP into people's view and hope our work could bring more attention to these problems which are common encountered in practical applications along with great potential values.
>
> * The Encoder-Decoder structure to deal with COPs are mainly end2end sequential models, they may  not catch the cluster information of the items. Our framework make good use of the connection matrix as a label to represent the cluster information of the bin packing problem, and it can be applied to other packing problems where the results are in the form of aggregation.
>
> * The Encoder and Decoder mentioned in our framework are separated. The Encoder can better characterize the characteristics of the problem itself, while the Decoder can modify the solution at instance-wise level. In actual application scenarios, some personalized or customized constraints can be solved using the mask mechanism combined with the Cluster Decoder.
>
> **Q3: Can you plot the gap vs time for different methods?**
>
> For all the baselines, only the heuristic methods ACO and GA are iterative algorithms during testing, the results of other algorithms do not change over time. We list several time v.s. the gap in the following table when $Q = 10, C = 5$.
>
> | Method   |   Time=20s |  Time=40s |   Time=60s |  Time=80s| Time=100s|
> |:----------:|:------:|:----:|:------:|:----:|:------:|
> |Random    |  5.03%|  5.03%|  5.03%| 5.03%|  5.03%|
> |FFD    |  5.00%|  5.00%|  5.00%| 5.00%|  5.00%|
> |GA    |  2.90%| 2.38%| 2.22%|2.10%| 2.05%|
> |ACO  |   4.64%| 4.40%|4.175%| 4.10%|3.90%|
> |Ptr  |   4.95%| 4.95%|4.95%| 4.95%|4.95%|
> |Ours|2.27%| 2.125%|2.10%|2.075%| 2.00%|

---

> > ### Comment · Reviewer_Rj4P · 2023-11-23
> >
> > Thanks so much for the response. Regarding on the new time vs gap result, it is beyond my expectations to see “Random” method do not improve over time. Random search should have the ability to find better solutions given more random trails.
> >
> > Generally speaking this paper is on the borderline that makes no mistakes. I intend to keep my score of 6

---

> > > ### Author Response · Authors · 2023-11-23
> > > **Thanks for your replying!**
> > >
> > > **Dear Reviewer Rj4P:**
> > >
> > >    Thanks for replying our response! We rechecked the learning curve of random and found that we only listed the **random packing result** with sepecific random seed which is consistant with our paper since we conducted random packing only once in our experiment. After reading your responses, we recognized that we didn't conduct **random search**  which recorded the best solution during each iteration. We re-implemented the  **random search** experiment and found that the gap is 5%,  5%,  4.975%, 4.975%,  4.95% over Time=20s, 40s, 60s, 80s and 100s.
> > >
> > > A more precise result of Random packing and Random Search can be listed as :
> > >
> > > | Method   |   Time=20s |  Time=40s |   Time=60s |  Time=80s| Time=100s|
> > > |:----------:|:------:|:----:|:------:|:----:|:------:|
> > > |Random   Packing |  5.03%|  -|  -| -| -|
> > > |Random  Search  |  5.00%|  5.00%|  4.975%| 4.975%|  4.95%|
> > >
> > > Random search is indeed a good strategy to find better solutions given more random trails. However, the search space in our problem is very large.  Since the probability of searching for the optimal solution is very small, the curve of random search decreases very slowly.
> > >
> > > Thanks again for your constructive suggestions in improving our paper during this rebuttal.

---

> ### Author Response · Authors · 2023-11-21
> **Reminder of Reviewer's feedback**
>
> **Dear Reviewer Rj4P:**
>
> We respectfully remind you that it has been more than 3 days since we submitted our rebuttal. We would appreciate your feedback on whether our response has addressed your concerns.
>
> In response to your comments, we have answered your concerns and improved the paper in the following aspects:
>
> * We claimed the novelty of their proposed Encoder-Decoder model over existing works  from three aspects, and this part was added in the revision of the manuscript .
>
> * We provided our contributions to other BPPs from three aspects.
>
> * We provided the gap vs time for different methods in table.
>
> Thanks again for your valuable review. We are looking forward to your response and are happy to answer any future questions.

---

### Official Review · Reviewer_2rkj · 2023-11-04

**Soundness:** 3 good
**Presentation:** 3 good
**Contribution:** 3 good
**Rating:** 8
**Confidence:** 5

**Summary:**

1.In this paper author proposed a vector BPP variant called Class-Constrained Bin Packing Problem (CCBPP), dealing with items of both
classes and sizes, and the objective is to pack the items in the least amount of bins respecting the bin capacity and the number of different classes that it can hold.

**Strengths:**

1.The pipeline encoder and decoder is quite unique work.
2. Presented results are compared with recent state of art techniques in detail.
3.More scope for the future researchers.

**Weaknesses:**

No

**Questions:**

1.Justify the need of encoder architecture proposed by Fraughnaugh, 1997 used here what advantages it has in the proposed work implementation.
2.Math mentioned in the decoder archticture in figure 2 should be mentioned in detail.

---

> ### Author Response · Authors · 2023-11-18
> **Response to Reviewer 2rkj**
>
> **Dear Reviewer 2rkj:**
>
> We appreciate your positive recognition of our work，we will address your concerns as follows:
>
> **Q1：Justify the need of encoder architecture proposed by （Fraughnaugh, 1997）**
>
> The reference cited in section 4.2 Encoder (Fraughnaugh, 1997) is an introduction to graph theory. It is cited here to illustrate that the CCBPP we want to solve can form a graph structure according to the characteristics of the problem, so that it can be encoded with a graph neural network.
>
> [1] Kathryn Fraughnaugh. Introduction to graph theory. Networks, 30:73, 1997.
>
> **Q2：Math mentioned in the decoder archticture in figure.2 should be mentioned in detail.**
>
> There are two formulas in the Decoder of the Figure.2, $C(\pi) = -R(\pi)$ and $L(w)= E_{\pi}[(C(\pi) - b) \nabla \log p_{\theta}(\pi|w)] $.
>
> In section 4.3.2, we mentioned that reinforcement learning technique is leveraged to fine-tune our solution at instance-wise level, and the reward from the environment $R(\pi)$ is set as the occupied bins corresponding to each sequence decoded by cluster decode strategy, and the cost for each instance is the opposite number of the reward $C(\pi) = -R(\pi)$.
>
> $L(w)= E_{\pi}[(C(\pi) - b) \nabla \log p_{\theta}(\pi|w)] $ is the loss function of Policy Gradient [1] where $b$ denotes the average reward of the batch data and  $p_{\theta}$ is the probability of one generated sequence. Since the active search part is not the main contribution of our paper, more details of active search can be referenced to [2].
>
>
> [1] Ronald J.Williams. Simple statistical gradient-following algorithms for connectionist reinforcement learning. Machine Learning, 8:229–256, 2004.
>
> [2] Andr´e Hottung, Yeong-Dae Kwon, and Kevin Tierney. Efficient active search for combinatorial optimization problems, 2022.

---

### Official Review · Reviewer_HxmL · 2023-11-07

**Soundness:** 2 fair
**Presentation:** 3 good
**Contribution:** 2 fair
**Rating:** 6
**Confidence:** 4

**Summary:**

This paper studies the Class Constrained Bin Packing Problem (CCBPP), which is a typical example of the vector Bin Packing Problem. The authors propose an encoder to predict the connectivity probabilities of different items and a fine-tuned decoder to output the solution. Experiments demonstrate that the proposed method outperforms baselines on several benchmarks.

**Strengths:**

1.	To the best of my knowledge, this paper is the first learning-based method to address Class-Constrained Bin Packing Problems.
2.	Experiments demonstrate that the proposed method outperforms baselines on several benchmarks.

**Weaknesses:**

1. This paper studies a typical example of the vector Bin Packing Problem. The authors may want to explain the significance and generality of the studied problem in detail.
2. This paper proposes an encoder-decoder model to solve Class-Constrained Bin Packing Problems. However, many existing learning-based methods for solving combinatorial optimization problems are based on Encoder-Decoder Models as well [1, 2, 3]. The authors may want to explain the novelty of their proposed method over existing works in detail.
3. The authors propose to use a graph neural network to generate a connection heatmap for classifying items into different packs. However, the motivation of using a heatmap to classify items rather than directly learning a node classification model is unclear.
4. Discussion on related work of other learning-based approaches for bin packing problems is missing.
5. The baselines are insufficient. Although the authors compare their method against the pointer network, recent work has proposed several improved models of the pointer network [4, 5]. The authors may want to compare their method with these baselines.
6.  It would be more convincing if the authors could evaluate their method on large-scale benchmarks, such as bin packing problems with over 1,000 items.

[1] Vinyals, Oriol, Meire Fortunato, and Navdeep Jaitly. "Pointer networks." Advances in neural information processing systems 28 (2015).

[2] Li, D., et al. "Solving Packing Problems by Conditional Query Learning." International Conference on Learning Representations (ICLR) Conference 2020.

[3] Zhang, Jingwei, Bin Zi, and Xiaoyu Ge. "Attend2pack: Bin packing through deep reinforcement learning with attention." arXiv preprint arXiv:2107.04333 (2021).

[4] Kool, Wouter, Herke van Hoof, and Max Welling. "Attention, Learn to Solve Routing Problems!." International Conference on Learning Representations. 2018.

[5] Nazari, Mohammadreza, et al. "Reinforcement learning for solving the vehicle routing problem." Advances in neural information processing systems 31 (2018).

**Questions:**

1. What is the technical novelty of the proposed method?
2. What is the motivation of using the heatmap to classify items?
3. Can the authors evaluate the proposed method on large-scale benchmarks?

---

> ### Author Response · Authors · 2023-11-18
> **Response to Reviewer HxmL (1/3)**
>
> **Dear Reviewer HxmL:**
>
> We appreciate your insightful comments of our work. We will address your comments as follows:
>
> **Q1: the significance and generality of CCBPP**
>
> We are sorry we don't claim the significance and generality of CCBPP clearly in our paper,  more about the CCBPP will be added into the revision of the manuscript. As far as we're concerned, the points are listed as follows:
>
> 1. **In the academic view, CCBPP as well as vector BPP are fully studied beyond neural combinatorial optimization community.**
>
>     BPP can be divided into two main categories, geometric BPP (2D and 3D BPP), and vector BPP [1]. The difference between these two problems is whether we should consider the geometrical constraints.
>     In the vector BPP research field, there are plenty of works proposed to solve vector BPP with different constraints [1, 2, 3]. Among them, class-constrained BPP is a typical variants. These research mainly focus on approximation heuristics [4] or exact solutions [5]. Unfortunately, learning-based methods have not been applied in this field so far.
>
> 2. **In the application view, CCBPP is commonly encountered in BPP field with wide applications.**
>
>      In general, BPP can be viewed as a resource allocation problem with resources from two kinds, incompatible resources and compatible resources (shared resources). The vanilla vector BPP only considers incompatible resources. Actually, there are plenty of real-world applications corresponding to both kinds of resources. For example, an operation machine can only process 2 kinds of operation (compatible resource) with limited materials (incompatible resources) [6]. This is quite common in many real-world applications, *Automatic Scaling in Cloud Computing [7]*, *Data-Placement Problem in Video-on-Demand* [8], *Production Planning* [9], *Co-painting Problem* [10], *Steel Mill Slab Problem* [6] are all typical real-world applications of CCBPP.
>
> [1] Christensen, Henrik I. , et al. "Approximation and online algorithms for multidimensional bin packing: A survey." Computer Science Review 24.May(2017):63-79.
>
> [2] Epstein, Leah , Csanád Imreh, and A. Levin . "Class constrained bin packing revisited." THEORETICAL COMPUTER SCIENCE -AMSTERDAM- (2010).
>
> [3] Kellerer, Hans and Ulrich Pferschy. “Cardinality constrained bin‐packing problems.” Annals of Operations Research 92 (1999): 335-348.
>
> [4] Shachnai, Hadas and Tami Tamir. “Polynomial time approximation schemes for class‐constrained packing problems.” Journal of Scheduling 4 (2001): 313-338.
>
> [5] Borges, Yulle G. F. et al. “Exact algorithms for class-constrained packing problems.” Comput. Ind. Eng. 144 (2020): 106455.
>
> [6] Crévits, Igor et al. “A special case of Variable-Sized Bin Packing Problem with Color Constraints.” 2019 6th International Conference on Control, Decision and Information Technologies (CoDIT) (2019): 1150-1154.
>
> [7] Xiao, Zhen et al. “Automatic Scaling of Internet Applications for Cloud Computing Services.” IEEE Transactions on Computers 63 (2014): 1111-1123.
>
> [8] Kochetov, Yury A. and A. Kondakov. “VNS matheuristic for a bin packing problem with a color constraint.” Electron. Notes Discret. Math. 58 (2017): 39-46.
>
> [9] Jansen, Klaus et al. “Approximation Algorithms for Scheduling with Class Constraints.” Proceedings of the 32nd ACM Symposium on Parallelism in Algorithms and Architectures (2019): n. pag.
>
> [10] Peeters, Marc and Zeger Degraeve. “The Co-Printing Problem: A Packing Problem with a Color Constraint.” Oper. Res. 52 (2004): 623-638.

---

> ### Author Response · Authors · 2023-11-18
> **Response to Reviewer HxmL (2/3)**
>
> **Q2：the novelty of their proposed method over existing works, and the technical novelty of the proposed method.**
>
> We are very sorry we don't compare our encoder-decoder frameworks over others in our paper. There are many learning-based encoder-decoder structures to solve combinatorial optimization problems [1-3] including bin packing problems, they are all end-to-end sequential models which means the both the encoder and the decoder are learned via sequences. However, an optimal packing result always corresponds to numerous packing sequences since shuffling the order of items within each bin does not affect the packing solution but will directly alter the packing sequence. Thus, it brings complexity when relying solely on learning from sequential models.
>
> Overall, the novelty of our proposed encoder-decoder framework are from three aspects:
>
> * In the encoder part, we introduce the connection matrix as a label to train the heat-map so that it can bring richer information than a packing sequence.
>
> * The Cluster-Decoder is designed to generate instance-wise packing solutions from heat-map, which successfully closes the gap between a heat-map and a final packing solution.
>
> * Our encoder and decoder are separated so that it is more flexible when applied in actual scene. The encoder is able to capture more problem-specific information, and the decoder can fine-tune the final solution at instance-wise level.
>
> [1] Vinyals, Oriol, Meire Fortunato, and Navdeep Jaitly. "Pointer networks." Advances in neural information processing systems 28 (2015).
>
> [2] Li, D., et al. "Solving Packing Problems by Conditional Query Learning." International Conference on Learning Representations (ICLR) Conference 2020.
>
> [3] Zhang, Jingwei, Bin Zi, and Xiaoyu Ge. "Attend2pack: Bin packing through deep reinforcement learning with attention." arXiv preprint arXiv:2107.04333 (2021).
>
>
> **Q3: the motivation of using a heatmap, why not use node classification?**
>
> Learning-based packing problems often use sequential models to construct solutions and they can also get promising results. In this paper, we offer another perspective to introduce the connection matrix as a label to represent whether different items are packed together in the optimal result, which can better represent the cluster information in the packing problems. The results in our paper show that the learned heap-map can better obtain the relationship of different items.
>
> Node classification may not be suitable for packing problems since there is no difference between different bins, which means the results reflect the aggregation of different items rather than categories. Clustering Algorithm such as knn is also not recommended since the packing constraints may not fully be satisfied, as mentioned in section 4.3.1.
>
> **Q4: related work of other learning-based approaches for bin packing problems is missing.**
>
> We will add the corresponding related work to the revision of the manuscript.
>
> **Q5: the baselines are insufficient.**
>
> Thanks for reminding us the improved version of ptr! In [1], the encoder was changed to an attention structure while the decoder still used the rnn structure. In [2], both the encoder and decoder were changed to a transformer-styled multi-head attention structure.
>
> We conduct experiments on the modified ptr[1,2]. Results below show that three different methods[1,2,3] do not make a big difference on CCBPP, which shows the limitation of sequence-to-sequence model when solving our problem. The rnn or the attention network may  not be able  to catch the constraints of different items as GCN does, and the  ptr cannot obtain the cluster information of different items as the heat-map does, as we stated in the third paragraph in section 1.
>
> |Method| **Q = 10,**| **C = 3**  | |**Q = 20,**  | **C = 3**| |**Q = 30,** |**C = 3** | |
> |:------:|:---------:|:------:|:----:|:----------:|:-------------:|:------:|:----------:|:----------:|:------:|
> | | Bins|     Gap      |  Time |Bins|     Gap      |  Time |Bins|     Gap      |  Time |
> |PointNetGreedy|21.41|7.05%|0.0181|21.88|9.40%|0.0181|22.11|10.55%|0.019|
> |PointNet-Sampling|21.07|5.35%|0.4|21.03|5.15%|0.41|21.26|6.30%|0.625|
> |Nazari. Etc - Greedy|21.44|7.20%|0.0135|21.82|9.10%|0.0157|22.12|10.60%|0.0179|
> |Nazari. Etc - Sampling|21.06|5.30%|0.2068|21.03|5.15%|0.2353|21.24|6.20%|0.2826|
> |Kool. Etc - Greedy|21.3|6.50%|0.0136|21.68|8.40%|0.0155|22.04|10.20%|0.0185|
> |Kool. Etc - Sampling|21.03|5.15%|0.2794|21.03|5.15%|0.2454|21.1|5.50%|0.2994|
> |GCN-Cluster,AS(Ours)|20.74|3.70%|10.14|20.91|4.55%|10.52|20.98|4.90%|14.05|

---

> ### Author Response · Authors · 2023-11-18
> **Response to Reviewer HxmL (3/3)**
>
> [1] Nazari, Mohammadreza, et al. "Reinforcement learning for solving the vehicle routing problem." Advances in neural information processing systems 31 (2018).
>
> [2] Kool, Wouter, Herke van Hoof, and Max Welling. "Attention, Learn to Solve Routing Problems!." International Conference on Learning Representations. 2018.
>
> [3] Irwan Bello, Hieu Pham, Quoc V. Le, Mohammad Norouzi, and Samy Bengio. Neural combinatorial
> optimization with reinforcement learning. ArXiv, abs/1611.09940, 2017.
>
> **Q6：experiments on large-scale benchmarks**
>
> We conducted experiments on $N=300$ and $N=500$， as shown in the following tables. The results show that our proposed method still has advantages compared to other baselines, and the results of large-scale benchmarks will be added to the revision of our manuscript.
> Larger-scale experiments when $N > 1000$  are not conducted at present due to the time and resource limitations. It brings us the inspiration to solve large-scale CCBPP with lower time complexity，which can be left to the future work.  There are some excellent works in the direction of combinatorial optimization that transplant models trained on small-scale problems to solve large-scale ones [1-3]， which will benefit us a lot.
>
> * N = 300
>
> | Method  |  N = 300| Q = 10   | C = 5 |  N = 300| Q = 20   | C = 5|
> |:----------:|:-------------:|:------:|:----:|:-------------:|:------:|:----:|
> |    | Bins|     Gap      |  Time |Bins|     Gap      |  Time |
> |Random  |   31.25 |4.17% |0.0136| 31.07|3.57% |0.0134|
> |FFD |   31.05|3.50%| 0.0141|31.125 |3.82%| 0.0145|
> |GA    |   30.58| 1.93%|27.99|30.62|2.07%|29.753|
> |ACO  |  30.625| 2.08%| 163.87|30.595|1.98%|179.704|
> |PointNet-Greedy|  31.09| 3.63%| 0.034|31.19|3.97%|0.0305|
> |PointNet-Sampling| 31.00|3.33%| 0.3537|31|3.33%|0.5056|
> |GCN-Cluster,Greedy |31.00 |3.33%|0.0644|30.995|3.32%|0.0702|
> |GCN-Cluster,Sampling |30.925|3.08%|0.6250|30.93|3.10%|0.6911|
> |GCN-Cluster, AS(Ours)| **30.545**|**1.82%**| 21.0235|**30.57**|**1.90%**|20.16|
>
> * N = 500
>
> | Method  |  N = 500| Q = 10  | C = 5 |  N = 500| Q = 20   | C = 5|
> |:----------:|:-------------:|:------:|:----:|:-------------:|:------:|:----:|
> |    | Bins|     Gap      |  Time |Bins|     Gap      |  Time |
> |Random  |   51.86 |3.72% |0.0409| 51.95 |3.90% |0.0409|
> |FFD |   51.415 |2.83%| 0.0308|51.76 |3.52%|0.0315|
> |GA    |   50.90| 1.80%|72.65| 51.00| 2.00%|76.65|
> |ACO  |  50.97| 1.94%| 388.10| 51.00| 2.00%| 418.45|
> |PointNet-Greedy| 51.91| 3.82%| 0.0509| 52.06| 4.12%| 0.0509|
> |PointNet-Sampling| 51.22|2.44%| 0.7997|51.46|2.92%| 0.8793|
> |GCN-Cluster,Greedy |51.465 |2.93%|0.1300|51.6450 |3.29%|0.1388|
> |GCN-Cluster,Sampling |51.005|2.01%| 1.4969|51.02|2.04%| 1.5866|
> |GCN-Cluster, AS(Ours)| **50.84**|**1.68%**| 63.72|**50.99**|**1.98%**| 64.52|
>
>
> [1]	Li, S., Yan, Z., & Wu, C. (2021). Learning to Delegate for Large-scale Vehicle Routing. Neural Information Processing Systems.
>
> [2]	Fu, Zhang-Hua et al. “Generalize a Small Pre-trained Model to Arbitrarily Large TSP Instances.” ArXiv abs/2012.10658 (2020): n. pag
>
> [3]	Son, Jiwoo et al. “Meta-SAGE: Scale Meta-Learning Scheduled Adaptation with Guided Exploration for Mitigating Scale Shift on Combinatorial Optimization.” International Conference on Machine Learning (2023).

---

> ### Author Response · Authors · 2023-11-21
> **Reminder of Reviewer's feedback**
>
> **Dear Reviewer HxmL :**
>
> We respectfully remind you that it has been more than 3 days since we submitted our rebuttal. We would appreciate your feedback on whether our response has addressed your concerns.
>
> In response to your comments, we have answered your concerns and improved the paper in the following aspects:
>
> * We provided more introduction about CCBPP both **from academic view** and **from application view**, and this part was added to the appendix of our updated manuscript.
>
> * We claimed the novelty of their proposed Encoder-Decoder model over existing works  from three aspects, and this part was added in the revision of the manuscript .
>
> * We stated the motivation of using heat-map rather than node classification.
>
> * We added the related work of neural methods for BPP in the new updated manuscript.
>
> * We provided more experiment results including the **performances of advanced ptr** and **results on large benchmarks** on several settings.
>
> Thanks again for your valuable review. We are looking forward to your response and are happy to answer any future questions.

---

> > ### Comment · Reviewer_HxmL · 2023-11-22
> >
> > Thanks for the authors' response, which has addressed most of my concerns. Thus, I would like to raise my score to 6.

---

### Official Review · Reviewer_aXcu · 2023-11-08

**Soundness:** 4 excellent
**Presentation:** 3 good
**Contribution:** 3 good
**Rating:** 6
**Confidence:** 4

**Summary:**

Authors propose a new learning-based solver for Class-Contrained Bin Packing Problem (CCBPP). CCBPP is commonly applied to planning optimization problems. As a good solution requires clustering of items in the same class, existing sequential learning-based algorithms struggle. Authors propose a GNN-based model, which is pre-trained on synthetic data with ground truths. The model is further fine-tuned with policy gradient. In order to account for the fact that good solutions should cluster items in the same class, a decoding algorithm that prioritizes items clustered to existing open bin is proposed. Experimental results demonstrate that the proposed algorithm outperforms reasonable heuristics, typical population-based algorithms, and standard Pointer Networks.

**Strengths:**

Originality, Significance: While many of the learning-based combinatorial optimization methods employ sequential models, CCBPP is a problem sequential models may not be best suited. Authors demonstrate that indeed the proposed method, which accounts for the clustering of items in the heatmap, outperforms previous sequential methods. This is an original contribution that will help the research community to think differently. As the use of GNNs & heatmaps are common in other methods, innovations in this paper, such as cluster decoding would also easily transfer to other problems in the future work.

Quality: Authors follow best practices in the literature. First, they use GNN to model the interaction between variables, which has become standard in neural combinatorial optimization algorithms. Then, they use heatmap for decoding, which is becoming standard for TSP and related problems. The choice of supervised pre-training is also very well-suited, because difficult problems with known ground-truth shall be easily generated. The proposed method also makes a good improvement over Pointer Network baseline. Experiments cover both synthetic and real-world data.

Clarity: The main ideas of the paper is mostly straightforward to follow, although I had some questions.

**Weaknesses:**

While the paper makes some methodological contributions such as GNN modeling of the problem and cluster-aware decoding algorithms, the significance of these proposed methods are contingent on the significance of CCBPP problem. As I am not an expert on bin packing and related problems, it is difficult for me to evaluate the significance of CCBPP. Also, other attendees of this conference may feel similarly. When authors present the paper, attendees wouldn't be interested in the talk unless they are convinced of the usefulness of CCBPP. While authors discuss CCBPP applications in Section 3, I would encourage authors to elaborate more on their significance in order to convince this conference's audience.

**Questions:**

In equation (8), wouldn't it numerically more tractable to optimize the log probability $\sum_{i,j} \hat{p}_{i,j}  \cdot \log p_{i,j}$, rather than sum of probabilities as in (8)? I understand this has nice interpretation as modularity, but often, probabilities are harder to optimize with gradient descent than log-probabilities, because sigmoid has near zero gradient for most of its domain.

I was also not sure how Policy Gradient shall be applied on ClusterDecode. Authors say $p_\theta$ in Policy Gradient equation (Section 4.3.2) correspond to Cluster Decode, but according to Appendix C, Cluster Decode is mostly deterministic algorithm (other than the choice of the first item). Hence, conditioned on the choice of the first item, the probability will be 1 for chosen item at the state, and 0 for not chosen items.

---

> ### Author Response · Authors · 2023-11-18
> **Response to Reviewer aXcu(1/2)**
>
> **Dear Reviewer aXcu:**
>
> We appreciate your recognition of the novelty and effectiveness of our work.  We will address your concerns as follows:
>
> **Q1: More introduction of CCBPP**
>
> We are sorry the introduction of CCBPP is not enough in our paper, more about the CCBPP will be added into the revision of the manuscript. As far as we are concerned, the significance of CCBPP can be listed as follows:
>
> 1. **In the academic view, CCBPP as well as vector BPP are fully studied beyond neural combinatorial optimization community.**
>
>     BPP can be divided into two main categories, geometric BPP (2D and 3D BPP), and vector BPP [1]. The difference between these two problems is whether we should consider the geometrical constraints.
>     In the vector BPP research field, there are plenty of works proposed to solve vector BPP with different constraints [1, 2, 3]. Among them, class-constrained BPP is a typical variants. These research mainly focus on approximation heuristics [4] or exact solutions [5]. Unfortunately, learning-based methods have not been applied in this field so far.
>
> 2. **In the application view, CCBPP is commonly encountered in BPP field with wide applications.**
>
>     In general, BPP can be viewed as a resource allocation problem with resources from two kinds, incompatible resources and compatible resources (shared resources). The vanilla vector BPP only considers incompatible resources. Actually, there are plenty of real-world applications corresponding to both kinds of resources. For example, an operation machine can only process 2 kinds of operation (compatible resource) with limited materials (incompatible resources) [6]. This is quite common in many real-world applications, *Automatic Scaling in Cloud Computing [7]*, *Data-Placement Problem in Video-on-Demand* [8], *Production Planning* [9], *Co-painting Problem* [10], *Steel Mill Slab Problem* [6] are all typical real-world applications of CCBPP.
>
> 3. Our proposed method is not restricted on CCBPP, indeed our method provides an efficient way to handle vector BPP with multiple properties and complex constraints, which is seldom studied in neural combinatorial optimization field. Furthermore, we hope our work could bring more attention to these problems which are common encountered in practical applications along with great potential values.
>
>
> [1] Christensen, Henrik I. , et al. "Approximation and online algorithms for multidimensional bin packing: A survey." Computer Science Review 24.May(2017):63-79.
>
> [2] Epstein, Leah , Csanád Imreh, and A. Levin . "Class constrained bin packing revisited." THEORETICAL COMPUTER SCIENCE -AMSTERDAM- (2010).
>
> [3] Kellerer, Hans and Ulrich Pferschy. “Cardinality constrained bin‐packing problems.” Annals of Operations Research 92 (1999): 335-348.
>
> [4] Shachnai, Hadas and Tami Tamir. “Polynomial time approximation schemes for class‐constrained packing problems.” Journal of Scheduling 4 (2001): 313-338.
>
> [5] Borges, Yulle G. F. et al. “Exact algorithms for class-constrained packing problems.” Comput. Ind. Eng. 144 (2020): 106455.
>
> [6] Crévits, Igor et al. “A special case of Variable-Sized Bin Packing Problem with Color Constraints.” 2019 6th International Conference on Control, Decision and Information Technologies (CoDIT) (2019): 1150-1154.
>
> [7] Xiao, Zhen et al. “Automatic Scaling of Internet Applications for Cloud Computing Services.” IEEE Transactions on Computers 63 (2014): 1111-1123.
>
> [8] Kochetov, Yury A. and A. Kondakov. “VNS matheuristic for a bin packing problem with a color constraint.” Electron. Notes Discret. Math. 58 (2017): 39-46.
>
> [9] Jansen, Klaus et al. “Approximation Algorithms for Scheduling with Class Constraints.” Proceedings of the 32nd ACM Symposium on Parallelism in Algorithms and Architectures (2019): n. pag.
>
> [10] Peeters, Marc and Zeger Degraeve. “The Co-Printing Problem: A Packing Problem with a Color Constraint.” Oper. Res. 52 (2004): 623-638.
>
> **Q2: Can the loss $L_m$ be modified to be easier to optimize with gradient descent?**
>
> Thanks for your constructive advice! Optimizing  $\sum_{ij} \log p_{ij} \cdot \hat{p}_{ij}$
>
> will make the $L_m$ too small so that it will be harder to set the balance coefficient of  $L_{m}$  and  $L_{ce}$. Still ,we are really appreciated for reminding us the difficulty of optimizing $\sum_{ij} p_{ij} \cdot \hat{p}_{ij}$, it will benefit us for the future work on loss function design.

---

> ### Author Response · Authors · 2023-11-18
> **Response to Reviewer aXcu (2/2)**
>
> **Q3:  How to calculate $p_{\theta}$ in section 4.3.2?**
>
> We are very sorry that we don't make it clear in our paper.
> - $\pi$ represents the **solution** generated by cluster decode strategy.
> - $p_\theta$ is the **probability of generating a packing sequence** based on the heat-map according to the cluster decode strategy.
> - $p_{\theta}(a_t|s_t,w)$ in section 4.3.2 is determined by the maximum value of the sum of the connection strengths corresponding to all items in the latest opened bin, as mentioned in section 4.3.1.
>
> Once the heat-map is obtained, a packing solution $\pi$ can be decoded by the cluster decoder deterministically, with $p_\theta$ calculated by $\prod_{t=1}^{N}{p_{\theta}(a_t|s_t,w)}$. After obtaining this probability $p_\theta$, the Policy Gradient can be leveraged to adjust the the non-fixed parameters of network so that the output of the network (heat-map) will be changed. Then the cluster-decoder is leveraged based on this 'new' heat-map to generate a 'new' packing solution. This procedure is repeated iteratively until no significant improvement of the solution.
>
> Thanks again for reminding us the detail, it will help us enhance the precision of the manuscript.

---

> ### Author Response · Authors · 2023-11-21
> **Reminder of Reviewer's attention and feedback**
>
> **Dear Reviewer aXcu:**
>
> We respectfully remind you that it has been more than 3 days since we submitted our rebuttal. We would appreciate your feedback on whether our response has addressed your concerns.
>
> In response to your comments, we have answered your concerns and improved the paper in the following aspects:
>
> * We provided more introduction about CCBPP both **from academic view** and **from application view**.
>
> * We clarified that it would be hard to balance the coefficient $\lambda$ of the two losses $L_m$ and $L_{ce}$, so we still chose to optimize $\sum_{ij} p_{ij} \cdot \hat{p}_{ij}$
>
> * We clarified the calculation of $p_{\theta}$, and added modified this part in the revision of our manuscript.
>
> Thanks again for your valuable review. We are looking forward to your response and are happy to answer any future questions.

---

### Author Response · Authors · 2023-11-18
**Response to all reviewers**

We would like to thank all reviewers for their thoroughly and insightful suggestions to improve our paper. We would like to emphasis some aspects regarding the concerns raised by several reviewers as follows.

**The significance and generality of CCBPP**

* **In the academic view, CCBPP as well as vector BPP are fully studied beyond neural combinatorial optimization community.**

* **In the application view, CCBPP is commonly encountered in BPP field with wide applications.**

* Our proposed method is not restricted on CCBPP, indeed our method provides an efficient way to handle vector BPP with multiple properties and complex constraints, which is seldom studied in neural combinatorial optimization field. Furthermore, we hope our work could bring more attention to these problems which are common encountered in practical applications along with great potential values.


**Novelty of Our Encoder-Decoder Framework over other structures**

* In the Encoder part, we introduce the connection matrix as a label to train the heat-map so that it can bring richer information than a packing sequence.

* The Cluster-Decoder is designed to generate instance-wise packing solutions from heat-map, which successfully closes the gap between a heat-map and a final packing solution.

* Our encoder and decoder are separated so that it is more flexible when applied in actual scenes. The encoder is able to capture more problem-specific information, and the decoder can fine-tune the final solution at instance-wise level.

Base on all the suggestions, we have updated our manuscript. All the changes are written in red in the new pdf version for ease of inspection. We would like to summarize what we have added and modified below:

* More introduction of CCBPP is added in Appendix A.

* The novelty of Our Encoder-Decoder Framework over other structures is added in section 1.

* Related work of learning-based BPP is added in section 2.

* The detail of Cluster Decode is modified in section 4.3.

* Experiments of large-scale benchmarks of CCBPP is addded in Appendix E.4.

---

### Public Comment · ~Renan_Fernando_Franco_Da_Silva1 · 2024-07-18

Hello,

In sections 5 and 5.1, you say that the results obtained by Silva and Schouery (2023) are hard to reproduce and that the benchmarks used are unavailable online. However, our solver and instances are available online at: https://gitlab.com/renanfernandofranco/a-branch-and-cut-and-price-algorithm-for-cutting-stock-and-related-problems

Feel free to contact us if you have any problems running the solver. In the worst case, we can run our solver over the benchmark that you created.

Best Regards,
Renan Franco

---

### Meta-Review · Area_Chair_TBjf · 2023-12-06

**Metareview:**

Summary: The submission presents a neural solver for a type of bin packing problem. Unlike previous methods, which focus on geometric bin packing, the current submission considers a harder case of class-constrained bin packing. Experiments show that the proposed method obtains good empirical results.

+ The paper shows the strength of ML for a more complex problem than the one considered previusly.
+ The empirical results show the proposed method is accurate.

- The methodological advances are incremental. The main novelty is in the new problem considered.

**Justification For Why Not Higher Score:**

The problem considered may not be of sufficient interest to a broad audience.

The methodological advances are incremental in nature.

The paper also lacks clarity, which may hinder its accessibility.

**Justification For Why Not Lower Score:**

The problem setting studied is more difficult than previous methods, and the empirical evidence clearly highlights the strength of ML in solving it accurately.

---

### Decision · Program_Chairs · 2024-01-16

Accept (poster)